# Learning Pseudometric-based Action Representations for Offline Reinforcement Learning

## Abstract

Offline reinforcement learning is a promising approach for practical applications since it does not require interactions with real-world environments. However, existing offline RL methods only work well in environments with continuous or small discrete action spaces. In environments with large and discrete action spaces, such as recommender systems and dialogue systems, the performance of existing methods decreases drastically because they suffer from inaccurate value estimation for a large proportion of out-of-distribution (o.o.d.) actions. While recent works have demonstrated that online RL benefits from incorporating semantic information in action representations, unfortunately, they fail to learn reasonable relative distances between action representations, which is key to offline RL to reduce the influence of o.o.d. actions. This paper proposes an action representation learning framework for offline RL based on a pseudometric, which measures both the behavioral relation and the data-distributional relation between actions. We provide theoretical analysis on the continuity and the bounds of the expected Q-values using the learned action representations. Experimental results show that our methods significantly improve the performance of two typical offline RL methods in environments with large and discrete action spaces.

## 1 Introduction

Reinforcement learning (RL) approaches have been applied successfully in many decision-making tasks. In a conventional setting, RL agents learn policies through an online fashion, where they collect trial-and-error experiences directly from the environment to improve the current policy. This can be done when the environment can be easily established, and the cost of deploying new policies online is low (e.g., computer games). However, many real-world scenarios allow only offline access to the environment due to cost or safety concerns (Thomas, 2015), meaning that the algorithm cannot access the environment during the training phase. This setting is known as offline RL, or batch RL (Levine et al., 2020). Despite the success that offline RL has achieved in robotic control tasks (Fujimoto et al., 2019b) and some Atari games (Gulcehre et al., 2021), the applicability of offline RL is still limited in many practical scenarios where the action space is large and discrete, including recommender systems and dialogue systems.

Prior works on offline RL mainly focus on the setting that the action space is continuous or only consists of a few discrete actions. To address the issue of overestimating the values of o.o.d. actions, they usually constrain the learned policy to stay close to the data-generating policies (Fujimoto et al., 2019b; Kumar et al., 2019; Wu et al., 2019; Kumar et al., 2020; Kostrikov et al., 2021; Zhou et al., 2020). However, this family of algorithms might suffer from poor performance when there are a large number of discrete actions. Firstly, the value function hardly generalizes over the entire action space without proper action representations because the actions are isolated in nature. By contrast, actions can be naturally represented in continuous control tasks by their physical semantics (e.g., directions, forces); thus, the value functions could generalize well. Secondly, with the size of the action space increasing, the state-action pairs are more sparse to the entire state-action space, resulting in a large proportion of o.o.d. actions. Consequently, the learned policies would be overly restrictive – they are constrained to only select actions within the support of in-distribution actions, without considering a large number of other o.o.d. actions that might contain the optimal action (Zhou et al., 2020; Kumar et al., 2019).

In this paper, we propose a novel framework for learning Behavioral Metric of Actions (**BMA**) to accelerate the offline RL tasks with large discrete action spaces. In BMA, the policy of an agent is trained with a latent, state-conditional action space, where the action representations are learned under a pseudometric that reflects

their behavioral and data-distributional similarities. The behavioral metric, inspired by the bisimulation metric defined on states (Zhang et al., 2021), aiming to explicitly quantify how similar that two actions' effects are on the environment. We also define a data-distributional metric, aiming to quantify how an action deviates from those actions in the dataset. Then, we propose an architecture of action encoder to learn action representations in a self-supervised way, where $\ell_1$ distances between representations correspond to the defined metric. The learned action representations can be combined with any offline RL algorithms and improve them in two ways. On the one hand, benefiting from the behavioral similarity between actions, the Q-function would be more smooth on the action space and, therefore, easier to converge. On the other hand, the quantified distributional distance enables the algorithm to balance between penalizing and exploring the o.o.d. actions.

We theoretically prove the continuity and the bound of the Q-values using our proposed action representations. We also empirically evaluate our methods on two simulated tasks and two real-world applications with large action sets. Experimental results show that policies trained under the BMA framework significantly outperform those trained using original actions. Our pseudometric-based action representations are substantially more effective to offline RL tasks when compared to prior action representations widely used in online RL tasks.

## 2 RELATED WORKS

**Offline RL** Offline RL aims to learn policies from logged transition data without any active data collection (Levine et al., 2020). Recently, many works have been proposed in this area (Kumar et al., 2019; Wu et al., 2019; Kumar et al., 2020; Kostrikov et al., 2021; Fujimoto et al., 2019b). They empirically and theoretically reveal that the critical issue in offline RL is the overestimation error induced by o.o.d. actions. As a result, they propose a variety of behavioral regularizations in RL training that compel the learned policy to stay close to the offline data. These regularizations consist of incorporating some divergence regularization into the critic (Kumar et al., 2020; Nachum et al., 2019), policy divergence penalties (Kumar et al., 2019; Wu et al., 2019), and appropriate network initializations (Matsushima et al., 2021). These works show effective results in simulated tasks, like Mujoco (Todorov et al., 2012) and Atari games (Gulcehre et al., 2021). However, we empirically show that they fail in the real-world applications with large discrete actions, likely due to the difficulty of generalization over large action sets or the excessive restriction induced by a large proportion of o.o.d. actions.

**Action representations in RL** In online RL, continuous action representations are usually used to exploit underlying structures of large discrete action spaces, thereby accelerating policy optimization in large-scale real-world tasks, like recommender systems (Ie et al., 2019) and dialogue systems (Lee et al., 2019). In prior works, Lee et al. (2018) uses predefined action embeddings to represent discrete actions and utilize continuous policy gradients for policy optimization. Chandak et al. (2019) avoid predefined embeddings by linking action representations to state transitions. Tennenholtz and Mannor (2019) regard action trajectories as natural languages and thus learn action representations from trajectories of expert demonstrations. Wang et al. (2021) learn action representations that focus on accurate reconstruction of rewards and next observations. Though these works show good results in online RL tasks, we empirically show that their performance is extremely unstable when combined with offline RL algorithms. The reason might be that they build statistical-based representations from raw data and thus struggle in the data-poor problem in the offline settings, especially when the action spaces are large. Furthermore, they lack an explicit scheme to regularize distances between action representations, which we find are important to the behavioral regularizations of offline RL algorithms. Our method utilizes a relation network (Santoro et al., 2017) to enforce the distances between action representations to reflect their behavioral and data-distributional relations. This provides extra information for building action representations when the offline data is sparse in the entire state-action space and helps offline RL algorithms to derive more proper behavioral regularizations. Some other works use action representations as pre-training behavioral priors or primitives (Zhou et al., 2020; Ajay et al., 2021; Singh et al., 2021). They train policies on latent action spaces which are learned from past successful trials and thus avoid a serious distributional shift in offline RL. However, these works seriously rely on the offline dataset's quality; experience data with poor performance heavily limits the policy improvement. On the other hand, they focus on continuous control tasks without any specific design to address large discrete action spaces.

**Metrics in RL** A crucial principle to generalization in reinforcement learning is to assign similar predictions to similar states. A standard implementation is to use the similarity in an adaptive fashion and group states into clusters while preserving some desired properties. The fundamental assumption behind it is the existence of a metric characterizing the real-valued distance between states (Le Lan et al., 2021). A related concept is

bisimulation metrics that measure how "behaviorally similar" states are Ferns et al. (2004). Recently, Zhang et al. (2021) extend this concept by proposing a gradient-based method for learning a representation space with the properties of bisimulation metrics. In the field of offline RL, Dadashi et al. (2021) propose a pseudometric to measure the closeness of state-action pairs to the support of logged transitions, but does so directly, without learning a representation. These works mainly focus on metrics on states or state-action pairs but ignore relations between actions. We believe that our work is the first to define a behavioral and data-distributional metric on actions and show that action representations based on this metric are beneficial to offline policy improvements.

## 3 Preliminaries

We start by introducing notation in this work. We consider the underlying environment as a Markov decision process (MDP) with a discrete action space, represented by the tuple $\mathcal{M} = (\mathcal{S}, \mathcal{A}, \mathcal{P}, \mathcal{R}, \gamma)$. Here, $\mathcal{S}$ is the state space. $\mathcal{A}$ is a finite set of actions, called the action set, and $|\mathcal{A}|$ denotes the size of the action set. $\mathcal{P}$ and $\mathcal{R}$ are the transition function and the reward function, respectively, indicating that when the agent takes the action $a \in \mathcal{A}$ under the state $s \in \mathcal{S}$, the probability of transitioning from state $s$ to state $s' \in \mathcal{S}$ is $\mathcal{P}(s'|s, a)$, and there is a environmental reward $\mathcal{R}(s, a) \in \mathbb{R}$. The goal of the agent is to learn a good policy $a \sim \pi(s)$ that maximizes the expected cumulative discounted rewards: $\mathbb{E}_{\mathcal{P}}[\sum_{t=1}^{\infty}[\gamma^t \mathcal{R}(s_t, \pi(s_t))]]$.

In online RL, the agent usually learns policy from its interacting experience with the current environment. By contrast, our work focus on the offline setting, in which the agent cannot collect new experience data and learns policy from a static dataset $\mathcal{D} = \{(s, a, s', r)\}$ generated by some other policy. We call the policy that generates $\mathcal{D}$ the *behavioral* policy and denote it as $\pi_\beta(a|s)$.

## 4 Offline RL with BMA

In this section, we elaborate on BMA, our proposed method for leveraging pseudometric-based action representations to address offline RL tasks with large discrete action spaces. We begin by describing the paradigm of how to train and execute policies according to the learned action representations. Next, we discuss which properties of action representations are crucial to the performance of this framework and define a pseudometric function to measure these properties. Then, we propose an action encoder structure to learn action representations complying with the desired properties from the offline dataset. Finally, we give a theoretical analysis of the generalization ability and the bound of the value function based on the learned action representation space.

**Overview.** We first introduce a two-phase paradigm for solving such tasks. It first learns an action representation space from the experience dataset by a self-supervised learning framework, and then, train an internal policy $\pi_i$ on the action representation space. This internal policy can build upon arbitrary offline RL algorithms with continuous control. In detail, we first train an action eocoder $\phi$ for generating action representations and then convert the original dataset to a new dataset $\mathcal{D}_e = \{(s, e, s', r)\}$, where $e = \phi(a; s)$ is the state-conditional action representation and $e \in \mathcal{E}$. Note that we introduce how to train $\phi$ in next subsections. Then, the applied offline RL algorithm would learn an interal policy $\pi_i(\hat{e}|s)$ from $\mathcal{D}_e$. It provides a latent action $\hat{e} \in \mathcal{E}$ for a given state, but $\hat{e}$ would likely not be a valid action, i.e., it does not equal any action representation $e$ of $a \in \mathcal{A}$. Therefore, we need to map from $\hat{e}$ to an element in $\mathcal{A}$. Here, we adopt a simple nearest neighbor lookup $g(\hat{e})$ introduced in Lee et al. (2018):

$$g(\hat{e}) = \arg\min_{a \in \mathcal{A}} \|\hat{e} - \phi(a; s)\|_1 \tag{1}$$

where $g$ is a mapping function from the continuous representation space to the discrete action set. It returns the original action whose representation is the closest to $\hat{e}$ by $\ell_1$ distance. Therefore, the overall policy $\pi_o = g(\pi_i(e|s))$. Its scheme is also described by Fig.1 (a) and Alg.1 in the appendix.

### 4.1 A pseudometric function for measuring relations between actions

In this section, we discuss which properties are important to our action representations and introduce a pseudometric function to measure these properties.

How to formulate the structure of action representations is crucial to the performance of our framework. The agent acting in the representation space would generalize the information of an action to other actions with

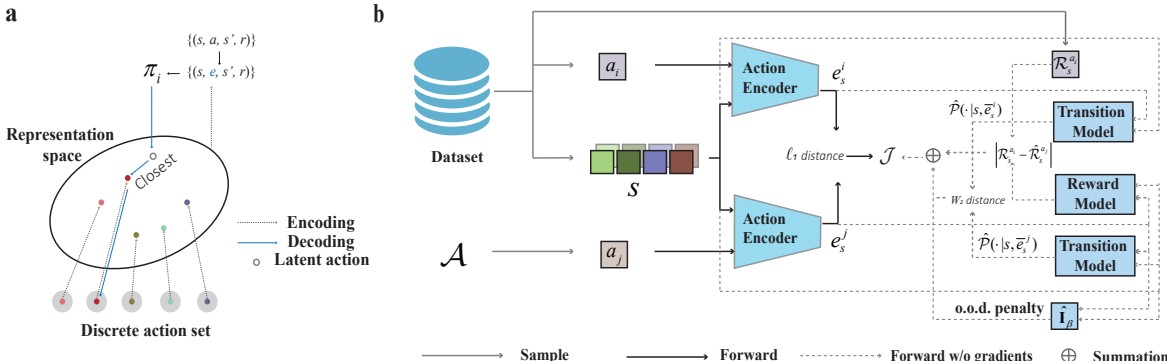

Figure 1: **BMA framework.** **(a)** The overall policy scheme. **(b)** The architecture of pseudometric-based action representation learning. The action encoder aims to generate action representations, whose $\ell_1$ distances between each other equals the expected pseudometric distance between their corresponding original actions. The generated action embeddings $e_s^i$ and $e_s^j$ are both conditioned on the same state $s$. During training, since it is hard to find two transitions that have the same state but different actions from the dataset, $(s, a_i, \mathcal{R}_s^{a_i})$ is sampled from the dataset, and $a_j$ is uniformly sampled from the finite action set $\mathcal{A}$. The reward of $(s, a_j)$ and the transitions of $(s, a_j)$ and $(s, a_i)$ are estimated by corresponding models. $\hat{I}_\beta$ is a function to estimate whether $a_j$ is an in-distribution action and thus determine whether there is an o.o.d. penalty on the pseudometric distance. The objective $\mathcal{J}$ is the mean square error between $\|e_s^i - e_s^j\|_1$ and the summation of the reward distance, the discounted transition distribution distance, and the o.o.d. penalty.

similar representations. However, if actions with similar representations have different environmental effects, this generalization might result in poor performance. Furthermore, existing offline RL algorithms usually adopt various approaches for behavioral regularizations to avoid serious overestimation errors; they enforce the learned policy to choose actions close to the in-distribution actions. Therefore, different action spaces induce distinct results of the behavioral regularizations, and an inappropriate representation space would destroy the effect of behavioral regularizations in our paradigm. For example. If the behaviorally different actions, which induce distinct transitions and rewards, are close in the representation space, a *false* result of behavioral regularization would happen. By contrast, the representation space where behaviorally similar actions are clustered helps the agent choose the optimal or safe actions. To this end, we expect that the learned action representations' relative distances reflect two major relations between any two discrete actions:

(i) **The behavioral relation**: The distance between any two actions in the representation space should reflect the difference between their induced transitions and rewards. This helps the behavioral regularized policy select actions close to the in-distribution actions in terms of their effects on the MDPs.

(ii) **The data-distributional relation**: The distance between any two actions in the representation space should reflect whether they are in the same distribution of the experience dataset or not. If they are not in the same distribution, there is a penalty distance between them. This helps offline RL algorithms tend to choose in-distribution actions, avoiding serious overestimation errors.

To derive the first relation, we first define a distance function $d : \mathcal{A} \times \mathcal{A} \mapsto \mathbb{R}_{\geq 0}$ measuring the behavioral similarity between two actions.

**Definition 1 (Behavioral Metric of Actions)** [1]*Given an MDP $\mathcal{M}$, a behavioral metric of actions is a function $d : \mathcal{A} \times \mathcal{A} \mapsto \mathbb{R}_{\geq 0}$ such that:*

$$d(a_i, a_j) = \mathbb{E}_{s \in \mathcal{S}} \left| \mathcal{R}_s^{a_i} - \mathcal{R}_s^{a_j} \right| + \gamma \cdot W_2(\mathcal{P}_s^{a_i}, \mathcal{P}_s^{a_j}) \tag{2}$$

*where $W_2$ is the $2^{nd}$ Wasserstein distance between two distributions.*

The Wasserstein metric is widely used in prior works (Zhang et al., 2021; Ferns et al., 2004) for measuring the distance between two probability distributions of states, indicating the cost of transporting mass from one

---

[1]Note that $d$ is a pseudometric, meaning that it allows the distance between two different actions to be 0.

distribution to another. Here, we adopt 2-Wasserstein metric $W_2$ in Eq.2, as it has a convenient formulation: $W_2(\mathcal{N}(\mu_i, \sum_i), \mathcal{N}(\mu_j, \sum_j))^2 = \|\mu_i - \mu_j\|_2^2 + \|\sum_i^{1/2} - \sum_j^{1/2}\|_\mathcal{F}^2$, where $\|\cdot\|_\mathcal{F}$ is the Frobenius norm.

Exactly computing the behavioral distance between two actions according to Eq.2 is generally impractical in the offline setting since the data points of states in the offline dataset are difficult to cover the entire state space, especially when the action space is large. Estimating the results of the max expectation over $\mathcal{S}$ might induce a large variance. Therefore, we extend the definition 1 to a conditional version as follows:

**Definition 2 (State-conditional Behavioral Metric of Actions)** *Given an MDP $\mathcal{M}$ and a state $s$, a state-conditional behavioral metric of actions is a function $d : \mathcal{S} \times \mathcal{A} \times \mathcal{A} \mapsto \mathbb{R}_{\geq 0}$ such that:*

$$d(a_i, a_j|s) = |\mathcal{R}_s^{a_i} - \mathcal{R}_s^{a_j}| + \gamma \cdot W_2(\mathcal{P}_s^{a_i}, \mathcal{P}_s^{a_j}) \tag{3}$$

This definition avoids estimating the expectation over $\mathcal{S}$ and brings a higher representation capability: It allows the actions' relations to be changed under different states.

**Data-distribution detection.** The major issue in offline RL is the overestimation problem caused by o.o.d. actions. Since o.o.d. actions are data-poor in the dataset, the estimation about relations of them might be inaccurate. This mismatch enables them to be close to those in-distribution actions which have distinct behavioral effects from them. So we also expect that there is a penalty distance between the o.o.d. actions and the in-distribution actions, forcing the behavioral regularized policy tends to choose in-distribution actions. Thus, we modify the pseudometric function $d$ (Eq.3) to reflect the data-distributional relation between actions:

$$d(a_i, a_j|s) = |\mathcal{R}_s^{a_i} - \mathcal{R}_s^{a_j}| + \gamma \cdot W_2(\mathcal{P}_s^{a_i}, \mathcal{P}_s^{a_j}) + p \cdot I_\beta(a_i, a_j|s) \tag{4}$$

where $p > 0$ is a penalty coefficient, $I_\beta(a_i, a_j|s)$ is a function indicating the relation of actions: If $a_i$ and $a_j$ are from two data distributions, $I_\beta(a_i, a_j|s) = 0$, otherwise, $I_\beta(a_i, a_j|s) = 1$.

## 4.2 LEARNING PSEUDOMETRIC-BASED ACTION REPRESENTATIONS

This section introduces how to learn action representations complying with the pseudometric $d$ (Eq.4) and gives a theoretical analysis about the generalization ability and the bound of the value function using our action representations.

In detail, we expect to train an action encoder $\phi : \mathcal{S} \times \mathcal{A} \mapsto \mathcal{E}$ for generating action representations that possess the desired property $d(a_i, a_j|s) := \|\phi(a_i; s) - \phi(a_j; s)\|_1$. It means that the $\ell_1$ distance between any two actions in the learned representation space equals their pseudometric distance. Concretely, the learning objective can be written as follows:

$$J(\phi) = \left(\|e_s^i - e_s^j\|_1 - d(a_i, a_j|s)\right)^2 \tag{5}$$

where $e_s^i = \phi(a_i; s), e_s^j = \phi(a_j; s)$. In practice, this objective requires sampling state-action pairs with the same state $s$ but different actions $a_i$ and $a_j$. However, it is challenging to search for sufficient such pairs from the offline dataset. Thus, we modify the objective by using the estimated samples:

$$J(\phi) = \mathbb{E}_{s, a_i, \mathcal{R}_s^{a_i} \sim \mathcal{D}, a_j \sim \mathcal{A}} \left(\|e_s^i - e_s^j\|_1 - \hat{d}(a_i, a_j|s)\right)^2 \tag{6}$$

where

$$\hat{d}(a_i, a_j|s) = |\mathcal{R}_s^{a_i} - \hat{\mathcal{R}}(s, \overline{e}_s^j)| + \gamma \cdot W_2(\hat{\mathcal{P}}(\cdot|s, \overline{e}_s^i), \hat{\mathcal{P}}(\cdot|s, \overline{e}_s^j)) + p \cdot \hat{I}_\beta(a_j|s) \tag{7}$$

Specifically, $\hat{d}(a_i, a_j|s)$ is an estimation of $d(a_i, a_j|s)$. $\overline{e}_s$ denotes $\phi(a; s)$ with stop gradients, $\hat{\mathcal{R}}$ and $\hat{\mathcal{P}}$ is the reward model and transition model, which have their own training steps. $\hat{I}_\beta(a_j|s)$ is also a trainable model to predict whether $a_j$ is an o.o.d. action. If $a_j$ is out of distribution, $\hat{I}_\beta(a_j|s) = 1$, otherwise $\hat{I}_\beta(a_j|s) = 0$. In practice, we follow the prior work to derive $\hat{I}_\beta(a|s)$ (Fujimoto et al., 2019a). We train a model $G(a|s) \approx \pi_\beta(a|s)$ to predict the probabilities of every action in the dataset and then scale all probabilities by the maximum probability. $a_j$ would be predicted as the o.o.d. action if the relative probability is below a threshold constant $\tau$. This procedure can be summarized as $\hat{I}_\beta(a|s) = \frac{G(a|s)}{\max\limits_{\hat{a} \in \mathcal{A}} G(\hat{a}|s)} \leq \tau$ .

During training, we sample the tuple $(s, a_i, \mathcal{R}_s^{a_i})$ from $\mathcal{D}$ and directly sample the other action $a_j$ from the finite action set $\mathcal{A}$. Then, other lacked data would be estimated by the trained models and the modified objective can be computed. To a certain extent, this estimation operation would reduce the accuracy of the learned action representations, but our empirical study shows that it is sufficient to derive a latent space with effective representative ability in offline RL tasks. Further details of training the action encoder and all models are described in Fig.1 () and Alg. 2 in the appendix.

**Theoretical Analysis.** We first theoretically analyze the generalization ability of the value function using our action representations. We prove that the value function of any given policy $\pi$ is Lipschitz with respect to our proposed pseudometric function $d$.

**Theorem 1** ($Q^\pi$ **is Lipschiz with respect to** $d$) *Given a policy $\pi$, let $Q^\pi$ be the value function for a given discount factor $\gamma$. $Q^\pi$ is Lipschitz continuous with respect to $d$ with a Lipschitz constant $\frac{1}{1-\gamma}$*

$$|Q^\pi(s, a_i) - Q^\pi(s, a_j)| \leq \frac{1}{1-\gamma} d(a_i, a_j | s) \tag{8}$$

Proofs in appendix. This theorem means that, the closer two actions are in terms of $d$, the more likely they are to share the similar value. If we explicitly force the distance between action representations to comply with $d$, the value function of the learned policy would be Lipschitz continuous in the action representation space. This continuity brings an effective generalization capability to the value function, and thus, intuitively, reduces the estimation errors of o.o.d. actions (Le Lan et al., 2021). On the other hand, when the policy is regularized to select actions close to the in-distribution actions in our representation space, it would choose similar actions in terms of the long-term return, avoiding selecting actions with distinct effects. Further, This conculsion can be further represented as: If one of $a_i$ and $a_j$ is the o.o.d. action but the other is not, $|Q^\pi(s, a_i) - Q^\pi(s, a_j)| \leq \frac{1}{1-\gamma} \cdot (|\mathcal{R}(s, a_i) - \mathcal{R}(s, a_j)| + \gamma \cdot W_2(\mathcal{P}(s'|s, a_i), \mathcal{P}(s'|s, a_j) + p)$, otherwise, $|Q^\pi(s, a_i) - Q^\pi(s, a_j)| \leq \frac{1}{1-\gamma} \cdot (|\mathcal{R}(s, a_i) - \mathcal{R}(s, a_j)| + \gamma \cdot W_2(\mathcal{P}(s'|s, a_i), \mathcal{P}(s'|s, a_j))$. This indicates that if two actions are from two distributions, the difference between their values tend to have a looser upper bound depending on the constant $p$, and $p$ plays a role in balancing between penalizing and exploring the o.o.d. actions. If $p$ is big, the difference between two values tends to be large, so the offline algorithms would ignore most o.o.d. actions. If $p$ is proper and the two actions' behavioral relation is close, the values o.o.d. actions and the values of in-distribution actions would be similar but avoid to be the same. This can encourage the offline policy to cautiously explore the effect of o.o.d. actions.

Now, we would show how action representations complying the pseudemetric $d$ improve downstream offline RL. Assuming that the BMA action encoder $\phi(a; s)$ has a learning error $\epsilon$: If the pseudometric distance between any two action is smaller than $\epsilon$, they would be aggregated together. Building on the policy theoretical analysis from (Ajay et al., 2021) and (Kumar et al., 2020), we now bound the performance of the policy obtained when offline RL is performed with BMA.

**Theorem 2 (Performance bound in offline RL.)** *Let $\pi_i^*(e|s)$ be the policy obtained by CQL performing with BMA in the constructed MDP $\overline{\mathcal{M}}$ and $\pi_{i,g}^*(a|s)$ refer to the overall policy when $\pi_i^*(e|s)$ is used together with nearest lookup function $g$. Let $\mathcal{J}(\pi, \mathcal{M})$ refer to the expected return of $\pi$ in $\mathcal{M}$ and $\phi(a; s)$ is the BMA action encoder, which has a learning error $\epsilon$. Let $\pi_\beta$ refer to the behavioral policy geanerting $\mathcal{D}$ and $\overline{\pi}_\beta(e|s) \equiv e = \phi(a; s), a \sim \pi_\beta(a|s)$. Then, $J(\pi_{i,g}^*, \mathcal{M}) \geq J(\pi_\beta, \mathcal{M}) - k$ where*

$$k = \mathcal{O}\left(\frac{1}{(1-\gamma)^2} \mathbb{E}_{s \sim d_{\overline{\mathcal{M}}}^{\pi_i^*(s)}} \left[\sqrt{|\mathcal{E}| D_{CQL}(\pi_i^*, \overline{\pi}_\beta)(s) + 1}\right]\right)$$
$$- \frac{\alpha}{1-\gamma} \mathbb{E}_{s \sim d_{\overline{\mathcal{M}}}^{\pi_i^*(s)}} [D_{CQL}(\pi_i^*, \overline{\pi}_\beta)(s)] + \frac{\epsilon}{1-\gamma} \tag{9}$$

Proofs in appendix. This bound suggests that the lower bound over the performance of the learned overall policy depends on three factors: 1) the divergence between the learned latent-space policy and the latent behavioral policy ($D_{CQL}(\pi_i^*, \overline{\pi}_\beta)(s)$), which is controlled by the applied offline algorithm. 2) The size of the latent space $|\mathcal{E}|$. According to Eq.(6) in (Kumar et al., 2020), note that if we directly learn an offline policy from the original action space $\mathcal{A}$, the lower bound of offline policy performance would depend on $|\mathcal{A}|$. This indicates that a latent

action space would induce an improvement on the worst-case deterioration over the learned policy, since we can set $|\mathcal{E}| \ll |\mathcal{A}|$, especially when $|\mathcal{A}|$ is enormous. 3) The learning error of BMA action encoder $\epsilon$. This term is induced by a self-supervised loss function. Compared with errors induced by offline RL algorithms, this term is much easier to be controlled. In conclusion, BMA provides a therotically gurrantee of policy improvement for offline RL task with large discrete action spaces.

## 5 EXPERIMENTS

In this section, we empirically show that BMA could be used as a drop-in extension for improving the policy performance in existing offline RL algorithms for problems with large discrete action spaces. Combing BMA with two standard offline RL algorithms, we evaluate our method in two simulated tasks and two real-world problems. All experiments in this paper are carried out with 5 different random seeds, and results are shown with a 95% confidence interval of standard deviation. We believe that the reported results can be further improved by using our framework with other offline RL algorithms; we leave this for future work. Details of architectures and hyperparameters can be found in the appendix.

**Evaluation Environments.** We consider four environments with large discrete action spaces: A **maze** environment introduced in Chandak et al. (2019). It has 4096 discrete actions. A **multi-step maze** environment that requires select actions from the 6-step planning decisions. It is modified from the maze environment and also has 4096 actions. A video **recommender system** that sets 1000 videos as the action set (Ie et al., 2019). A **dialogue system** that sets 1500 dialogues as the action set. The experience trajectories are policies with sub-optimal performance. They are all trained by online methods. Note that more details of environments and datasets can be found in the appendix.

### 5.1 EXPERIMENTAL RESULTS

We empirically reveal that optimizing policies on the learned action representation spaces leads to better performance than directly training policies on the original action spaces. We consider two widely-used offline RL algorithms: CQL (Kumar et al., 2020) and BCQ (Fujimoto et al., 2019b). We first learn our pseudometric-based action representations from the given datasets. Then, we utilize our framework to train policies using the two offline RL algorithms. The developed policies are annotated by **BMA-CQL** and **BMA-CQL**, respectively. The baselines are the **discrete** versions of **CQL** and **BCQ** and behavior cloning (**BC**).

The average performance curves are illustrated in Fig.2 (a). We also visualize the average performance of the **behavioral** policy. Overall, we find that BMA-CQL and BMA-BCQ show faster convergence and better performance than their discrete versions in the four environments. Furthermore, discrete CQL and discrete BCQ cannot outperform the behavior policies and show unstable learning curves in some situations. This indicates that in offline RL with large action sets, directly training policies on the original action spaces might suffer from serious generalization or extrapolation errors. In addition, the poor performance of BC in most environments indicates that simple imitation learning cannot address this problem well.

**Comparison with Other Action Encoders.** A natural question is whether training policies on other kinds of action representation spaces can lead to similar results? To reply to this question, we compare BMA-CQL against baselines using the same architecture but exchange our action representations with other representations. We first consider two random action representations to examine whether simply projecting discrete actions into continuous action spaces can improve the performance. The first scheme generates random vectors ranging from -1 to 1 as action representations (**Rand-1**). The other scheme also generates random vectors but forces the $\ell_1$ distance between any two action embeddings to be larger than a constant. This scheme avoids actions overlapping together (**Rand-2**). Then, we consider three schemes for learning action representations widely used in online tasks: (1) **Transition** (Chandak et al., 2019): An encoder that captures information about corresponding state transitions into action representations. (2) **Reconstruction**: An encoder-decoder scheme in which the decoder takes the state and the action representation generated by the encoder as input, and its objective is to predict the next state and reward. (3) **External** (Lee et al., 2018): Action representations are given by external information. In the maze environment, the action embeddings are set as the movement vectors. In the multi-step environment, the actions are represented as the concatenation of the base action at each step of the plan. In the recommendation system, the action embedding is the concatenation of video type (expressed as a one-hot vector), video quality, and video length. Note that there is no proper external information to represent dialogue actions, so we did not

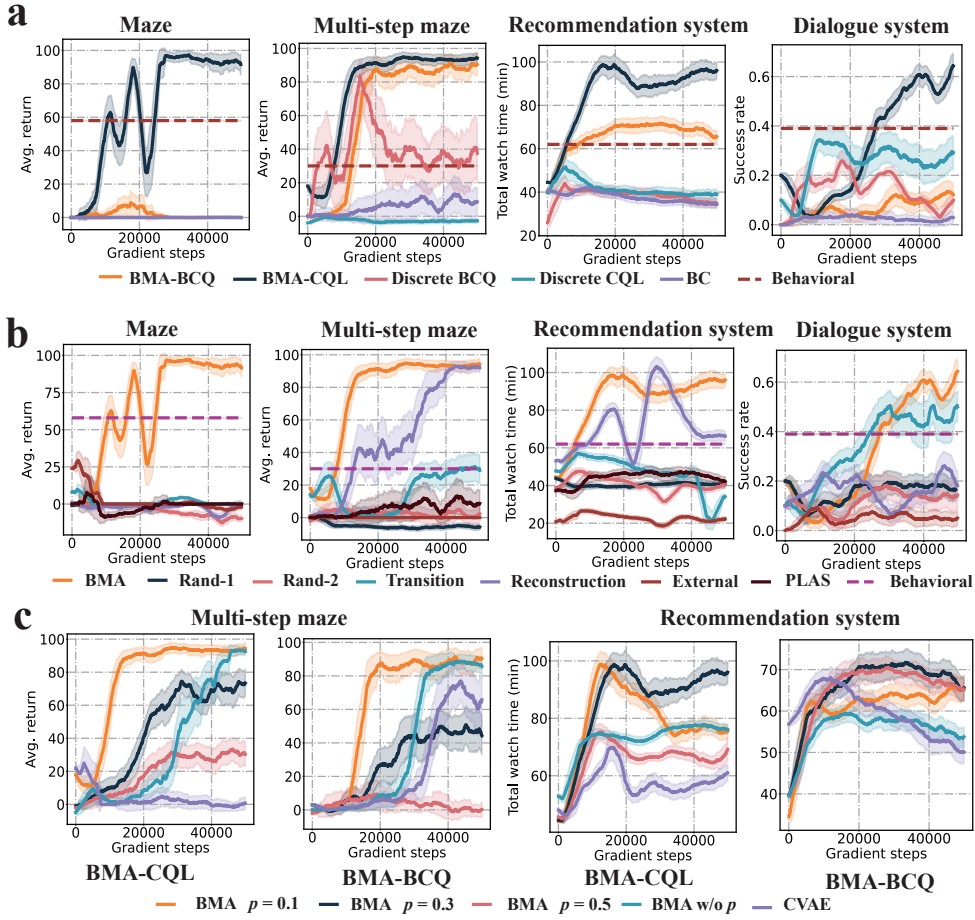

Figure 2: **Experimental results.** **(a)** Comparing BMA equipped with BCQ and CQL against directly training offline RL algorithms (Discrete BCQ, Discrete CQL, and BC) on the original action spaces in 4 environments with large action spaces. **(b)** Comparing the performance of BMA against other widely used action representations (transition-based representation, reconstruction-based representation, two kinds of random representations, and external representation ) in 4 environments with large action spaces. **(c)** Ablations. We perform two offline algorithms (CQL and BCQ) with BMA in two environments (Multi-step maze and Recommendation system). We consider BMA with different penalty distances (BMA with $p = 0.1, 0.3, 0.5$), removing the penalty distance from the learning objective (BMA w/o p), and removing the distance learning objective (Eq.6) from the learning procedure (CVAE).

consider external action representations in this task. Finally, we consider a similar work for offline RL (**PLAS** (Zhou et al., 2020)) since it also trains policies on the latent action spaces.

The average performance curves in four environments are illustrated in Fig.2 (b). Two random action representations both fail in the four environments, indicating that simply converting discrete action spaces to continuous action spaces cannot address this task. By comparison, transition and reconstruction representations show better performance in certain situations, revealing that generalization over actions with similar effects can improve the policy performance. However, these two representations also show poor performance in most environments. In contrast, our methods show superior events among all environments, revealing that action representations complying with the proposed pseudometric are more suitable to offline RL. In addition, external representations and PLAS both show poor performance in all tasks. The reason might be that external representations do not capture the information of dynamics in MDPs, resulting in misleading generalization effects. The objective of PLAS is to constrain policies more naturally instead of improving the generalization ability of action representations, so it

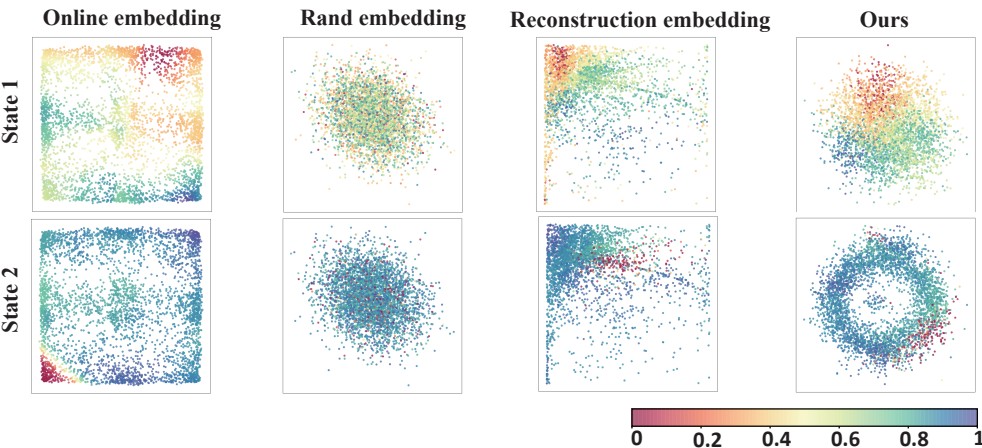

Figure 3: **Visualization of different representation spaces.** Since we set the dimension of representations as 2 in the multi-step maze environment, we directly plot the action representations. The position of each dot in these plots equals the corresponding action embeddings, and the color corresponds to the ground-truth action value predicted by a model (Chandak et al., 2019) with online training. Two rows are in terms of two different states. Each column from left to right corresponds to the representations learned by the online model, the rand-1 representations, the reconstruction representations, and ours.

cannot address the tasks with large discrete action spaces well. Fig.3 illustrates the action representations. We sample two states from the multi-step maze environment and plot the different action representations. We see that, in offline settings, the rand action representations cannot generalize well since actions with different values are mixed together. The reconstruction-based representations and our pseudometric-based representations both show generalization over actions: The changes of action value are continuous in both representation spaces to a certain extent. Furthermore, the reconstruction-based representations are scattered throughout the space while our representations with similar values are arranged as a more compact structure. Intuitively, we believe that this is the reason why our method shows faster convergence performance than reconstruction-based representations.

**Ablations** We perform several ablations on the multi-step maze environment and recommendation system to try and determine the importance of each component in our framework. Fig.2 (c) illustrate the results. We first consider the effects of different scales of the penalty constant. We set $p = 0.1, 0.3, 0.5$ respectively. The results show that a proper penalty constant would bring better performance, but a large penalty constant might make the performance collapse in some situations. The reason might be that large penalty distances destroy the generalization structure of action representations. Then, we consider removing the penalty constant (denoted as **BMA w/o p**). The results of this ablation show that without the distance penalty mechanism to ensure the distribution detection, there is performance degradation in our framework. Finally, we remove the objective function (Eq.6) for controlling the distances between actions. As a result, the architecture of this scenario is similar to a conditional variational auto-encoder that takes current states and actions as input and aims to reconstruct next states and rewards (denoted as **CVAE**) from the action embeddings. This ablation shows worse performance than **BMA w/o p** in many situations, indicating that controlling the pseudometric distances between actions provides extra information for action representations, leading to better policy performance.

## 5.2 CONCLUSIONS

This paper proposes Behavioral Metrics of Actions: a new action representation learning framework for offline RL tasks with large discrete action spaces. In the learned action representation spaces, distances between representations of discrete actions reflect their behavioral and data-distributional relations. We derive theoretically statements about the benefits of learning policies based on our action representations. Experimental results show that our methods significantly outperform prior works in a variety of environments.

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

## A APPENDIX

### A.1 ADDITIONAL THEOREMS AND PROOFS

**Theorem 1 ($Q^\pi$ is Lipschiz with respect to** $d$**)** *Given a policy $\pi(a|s)$, let $Q^\pi$ be the value function for a given discount factor $\gamma$. $Q^\pi$ is Lipschitz continuous with respect to $d$ with a Lipschitz constant $\frac{1}{1-\gamma}$*

$$|Q^\pi(s, a_i) - Q^\pi(s, a_j)| \leq \frac{1}{1-\gamma} d(a_i, a_j|s) \tag{10}$$

*Proofs.* follows by expanding each $Q$, rearranging terms and then simplifying the expression.

$$
\begin{aligned}
|Q^\pi(s, a_i) - Q^\pi(s, a_j)| &= |\mathcal{R}(s, a_i) + \gamma \cdot \sum_{s'} \mathcal{P}(s'|s, a_i) V^\pi(s') - (\mathcal{R}(s, a_j) + \gamma \cdot \sum_{s'} \mathcal{P}(s'|s, a_j)) V^\pi(s')| \\
&\leq |\mathcal{R}(s, a_i) - \mathcal{R}(s, a_j)| + \gamma \cdot \sum_{s'} |\mathcal{P}(s'|s, a_i) - \mathcal{P}(s'|s, a_j)| \cdot V^\pi(s')
\end{aligned}
\tag{11}
$$

We assume that $\mathcal{R} \leq 1$, so we get[2]

$$V^\pi(s) = \sum_{t=0}^{\infty} \gamma^t \cdot \mathcal{R}_t \leq \frac{1}{1-\gamma} \tag{12}$$

Combing the above equations, we have

$$
\begin{aligned}
|Q^\pi(s, a_i) - Q^\pi(s, a_j)| &\leq |\mathcal{R}(s, a_i) - \mathcal{R}(s, a_j)| + \gamma \cdot \sum_{s'} |\mathcal{P}(s'|s, a_i) - \mathcal{P}(s'|s, a_j)| \cdot V^\pi(s') \\
&\leq |\mathcal{R}(s, a_i) - \mathcal{R}(s, a_j)| + \frac{\gamma}{1-\gamma} \cdot \sum_{s'} |\mathcal{P}(s'|s, a_i) - \mathcal{P}(s'|s, a_j)| \\
&= \frac{1}{1-\gamma} \cdot \left( (1-\gamma) \cdot |\mathcal{R}(s, a_i) - \mathcal{R}(s, a_j)| + \gamma \cdot \sum_{s'} |\mathcal{P}(s'|s, a_i) - \mathcal{P}(s'|s, a_j)| \right) \\
&\leq \frac{1}{1-\gamma} \cdot \left( |\mathcal{R}(s, a_i) - \mathcal{R}(s, a_j)| + \gamma \cdot \sum_{s'} |\mathcal{P}(s'|s, a_i) - \mathcal{P}(s'|s, a_j)| \right)
\end{aligned}
\tag{13}
$$

---

[2]In practice, there might be some environments in which $\mathcal{R} \in [-\mathcal{R}_{max}, \mathcal{R}_{max}]$ and $\mathcal{R}_{max} > 1$. To address this issue, we normalize the reward signals by setting $\mathcal{R} \leftarrow \mathcal{R}/\mathcal{R}_{max}$

Since in Eq.(4), $p > 0$ and $I_\beta(a_i, a_j|s) \in \{0, 1\}$, we get

$$
\begin{aligned}
|Q^\pi(s, a_i) - Q^\pi(s, a_j)| &\le \frac{1}{1-\gamma} \cdot \left( |\mathcal{R}(s, a_i) - \mathcal{R}(s, a_j)| + \gamma \cdot \sum_{s'} |\mathcal{P}(s'|s, a_i) - \mathcal{P}(s'|s, a_j)| + p \cdot I_\beta(a_i, a_j|s) \right) \\
&\le \frac{1}{1-\gamma} d(a_i, a_j|s)
\end{aligned}
\tag{14}
$$

According to theorem 1, we can derive the bound between value functions in the original MDP and the MDP constructed by the learned BMA action representations. Assuming that the BMA action encoder $\phi(a; s)$ has a learning error $\epsilon$: If the pseudometric distance between any two action is smaller than $\epsilon$, they would be aggregated together.

**Lemma 1 (Value bound based on $\phi$)** *Given the BMA action encoder $\phi(a; s)$ which has a learning error $\epsilon$ and maps from actions in the original MDP $\mathcal{M}$ to actions in the MDP $\overline{\mathcal{M}}$ constructed by the action representations. The value functions for a given policy $\pi$ in the orginal MDP and its converted version $\overline{\pi}$ in the constructed MDP are bounded as:*

$$
|Q^\pi_{\mathcal{M}}(s, a) - Q^{\overline{\pi}}_{\overline{\mathcal{M}}}(s, \phi(a; s))| \le \frac{\epsilon}{1-\gamma}
\tag{15}
$$

*Proofs.* First, let us give some notation and assumptions concerning the form of the constructed MDP $\overline{\mathcal{M}}$. We assume $\overline{\mathcal{M}} = (\mathcal{S}, \mathcal{A}_\phi, \mathcal{P}_\phi, \mathcal{R}_\phi, \gamma)$, where $\mathcal{S}$ is the same state space, $\mathcal{A}_\phi$ is the action representation space where the $\ell_1$ distances between any two representations corresponds to our defined pseudometric distance, and there exists a learning error of $\phi$: If the pseudometric distance between any two action is smaller than $\epsilon$, they would be aggregated together. Recall that the nearest neighbor lookup $g$ is as follow:

$$
g(e) = \arg\min_{a \in \mathcal{A}} \|e - \phi(a; s)\|_1
\tag{16}
$$

As a result, we can get the relationships between the transition probabilities and rewards in the two MDPs, i.e.

$$
\begin{aligned}
\mathcal{P}_\phi(s'|s, e) &= \mathcal{P}(s'|s, g(e)) \\
\mathcal{R}_\phi(s, e) &= \mathcal{R}(s, g(e))
\end{aligned}
\tag{17}
$$

By this way, we can get

$$
\begin{aligned}
&|Q^\pi_{\mathcal{M}}(s, a) - Q^{\overline{\pi}}_{\overline{\mathcal{M}}}(s, \phi(a; s))| \\
&= |\mathcal{R}(s, a) + \gamma \cdot \sum_{s'} \mathcal{P}(s'|s, a)V^\pi_{\mathcal{M}}(s') - (\mathcal{R}_\phi(s, \phi(a; s)) + \gamma \cdot \sum_{s'} \mathcal{P}_\phi(s'|s, \phi(a; s)))V^{\overline{\pi}}_{\overline{\mathcal{M}}}(s'))| \\
&= |\mathcal{R}(s, a) + \gamma \cdot \sum_{s'} \mathcal{P}(s'|s, a)V^\pi_{\mathcal{M}}(s') - (\mathcal{R}(s, g(\phi(a; s))) + \gamma \cdot \sum_{s'} \mathcal{P}(s'|s, g(\phi(a; s)))V^{\overline{\pi}}_{\overline{\mathcal{M}}}(s'))| \\
&\le |\mathcal{R}(s, a) - \mathcal{R}(s, g(\phi(a; s)))| + \gamma \cdot |\sum_{s'} \mathcal{P}(s'|s, a) - \sum_{s'} \mathcal{P}(s'|s, g(\phi(a; s)))|V^\pi_{\mathcal{M}}(s')
\end{aligned}
\tag{18}
$$

According to the proofs of theorem 1, we get

$$
\begin{aligned}
|Q^\pi_{\mathcal{M}}(s, a) - Q^{\overline{\pi}}_{\overline{\mathcal{M}}}(s, \phi(a; s))| &\le \frac{1}{1-\gamma} d(a, g(\phi(a; s))|s) \\
&\le \frac{\epsilon}{1-\gamma}
\end{aligned}
\tag{19}
$$

We have argued that there exists some near-optimal value functions based on the constructed MDP $\overline{\mathcal{M}}$, if $\phi$ is sufficiently learned. Now, we would show how action representations complying the pseudemetric $d$ improve downstream offline RL. Building on the policy theoretical analysis from (Ajay et al., 2021) and (Kumar et al., 2020), we now bound the performance of the policy obtained when offline RL is performed with BMA.

**Theorem 2 (Performance bound in offline RL.)** *Let $\pi^*_i(e|s)$ be the policy obtained by CQL performing with BMA in the constructed MDP $\overline{\mathcal{M}}$ and $\pi^*_{i,g}(a|s)$ refer to the overall policy when $\pi^*_i(e|s)$ is used together*

with nearest lookup function $g$. Let $\mathcal{J}(\pi, \mathcal{M})$ refer to the expected return of $\pi$ in $\mathcal{M}$ and $\phi(a; s)$ is the BMA action encoder, which has a learning error $\epsilon$. Let $\pi_\beta$ refer to the behavioral policy geneartng $\mathcal{D}$ and $\overline{\pi}_\beta(e|s) \equiv e = \phi(a; s), a \sim \pi_\beta(a|s)$.Then, $J(\pi_{i,g}^*, \mathcal{M}) \geq J(\pi_\beta, \mathcal{M}) - k$ where

$$
\begin{aligned}
k = &\mathcal{O}\left( \frac{1}{(1-\gamma)^2} \mathbb{E}_{s \sim d_{\overline{\mathcal{M}}}^{\pi_i^*(s)}} \left[ \sqrt{|\mathcal{E}| D_{CQL}(\pi_i^*, \overline{\pi}_\beta)(s) + 1} \right] \right) \\
&- \frac{\alpha}{1-\gamma} \mathbb{E}_{s \sim d_{\overline{\mathcal{M}}}^{\pi_i^*(s)}} [D_{CQL}(\pi_i^*, \overline{\pi}_\beta)(s)] + \frac{\epsilon}{1-\gamma}
\end{aligned}
\tag{20}
$$

*Proofs.* First, let's break $|J(\pi_{i,g}^*, \mathcal{M}) - J(\pi_\beta, \mathcal{M})|$ into

$$
\begin{aligned}
|J(\pi_{i,g}^*, \mathcal{M}) - J(\pi_\beta, \mathcal{M})| \leq\ & |J(\pi_{i,g}^*, \mathcal{M}) - J(\pi_i^*, \overline{\mathcal{M}})| \\
& + |J(\pi_i^*, \overline{\mathcal{M}}) - J(\overline{\pi}_\beta, \overline{\mathcal{M}})| \\
& + |J(\overline{\pi}_\beta, \overline{\mathcal{M}}) - J(\pi_\beta, \mathcal{M})|
\end{aligned}
\tag{21}
$$

where $\overline{\pi}_\beta$ is the converted behavioral policy executed on the constructed MDP $\overline{\mathcal{M}}$ mentioned in the proofs of thereom **??**. It can be defined as $e \sim \overline{\pi}_\beta \equiv e = \phi(a; s), a \sim \pi_\beta(a|s)$. According to theorem 3.6 in (Kumar et al., 2020), we apply it to $\overline{\mathcal{M}}$, we get

$$
\begin{aligned}
&|J(\pi_i^*, \overline{\mathcal{M}}) - J(\overline{\pi}_\beta, \overline{\mathcal{M}})| \\
&\leq 2\left( \frac{C_{r,\delta}}{1-\gamma} + \frac{\mathcal{P}_{\phi,\delta}}{(1-\gamma)^2} \right) \mathbb{E}_{s \sim d_{\overline{\mathcal{M}}}^{\pi_i^*(s)}} \left[ \sqrt{\frac{|\mathcal{E}|}{|\mathcal{D}(s)|} D_{CQL}(\pi_i^*, \overline{\pi}_\beta)(s) + 1} \right] \\
&- \frac{\alpha}{1-\gamma} \mathbb{E}_{s \sim d_{\overline{\mathcal{M}}}^{\pi_i^*(s)}} [D_{CQL}(\pi_i^*, \overline{\pi}_\beta)(s)] = k_1
\end{aligned}
\tag{22}
$$

Then, we try to bound $|J(\pi_{i,g}^*, \mathcal{M}) - J(\pi_i^*, \overline{\mathcal{M}})|$. Since the decisions of $\pi_i^*$ and $\pi_{i,g}^*$ are in one-to-one correspondence, we get $|J(\pi_{i,g}^*, \mathcal{M}) - J(\pi_i^*, \overline{\mathcal{M}})| = 0$.

Now, we try to bound $|J(\overline{\pi}_\beta, \overline{\mathcal{M}}) - J(\pi_\beta, \mathcal{M})|$:

$$
\begin{aligned}
|J(\overline{\pi}_\beta, \overline{\mathcal{M}}) - J(\pi_\beta, \mathcal{M})| &= |\sum_{s \sim \mathcal{S}} d_0(s) V_{\overline{\mathcal{M}}}^{\overline{\pi}_\beta}(s) - \sum_{s \sim \mathcal{S}} d_0(s) V_{\mathcal{M}}^{\pi_\beta}(s)| \\
&= |\sum_{s \sim \mathcal{S}} d_0(s) \sum_{e \sim \mathcal{E}} \overline{\pi}_\beta(e|s) Q_{\overline{\mathcal{M}}}^{\overline{\pi}_\beta}(s, e) - \sum_{s \sim \mathcal{S}} d_0(s) \sum_{a \sim \mathcal{A}} \pi_\beta(a|s) Q_{\mathcal{M}}^{\pi_\beta}(s, a)| \\
&= |\sum_{s \sim \mathcal{S}} d_0(s) \sum_{a \sim \mathcal{A}} \pi_\beta(a|s) Q_{\overline{\mathcal{M}}}^{\overline{\pi}_\beta}(s, e) - \sum_{s \sim \mathcal{S}} d_0(s) \sum_{a \sim \mathcal{A}} \pi_\beta(a|s) Q_{\mathcal{M}}^{\pi_\beta}(s, a)| \\
&= \sum_{s \sim \mathcal{S}} d_0(s) \sum_{a \sim \mathcal{A}} \pi_\beta(a|s) |Q_{\overline{\mathcal{M}}}^{\overline{\pi}_\beta}(s, e) - Q_{\mathcal{M}}^{\pi_\beta}(s, a)|
\end{aligned}
\tag{23}
$$

According to lemma 1, $|Q_{\overline{\mathcal{M}}}^{\overline{\pi}_\beta}(s, e_s^a) - Q_{\mathcal{M}}^{\pi_\beta}(s, a)| = \frac{\epsilon}{1-\gamma}$, so $|J(\overline{\pi}_\beta, \overline{\mathcal{M}}) - J(\pi_\beta, \mathcal{M})| = \frac{\epsilon}{1-\gamma} = k_2$. Finally, we get $k = k_1 + k_2$. We apply $\mathcal{O}$ to get the notation in the theorem.

## A.2 ALGORITHMS

## A.3 ARCHITECTURE, HYPERPARAMETERS, AND INFRASTRUCTURE

In our implementations, the nearest neighbor lookup function $g$ can be regarded as finding the minimum distance between $\hat{e}$ and all actions' embeddings. So, we directly adopted the `torch.min()` function in the PyTorch python package. Its time complexity is $\mathcal{O}(N)$, so there would not be a serious scale problem when the action set is enormous.

For $\hat{I}_\beta(a|s)$. We train a model $G(a|s) \approx \pi_\beta(a|s)$ to predict the probabilities of every action under a given state $s$ and then scale all probabilities by the maximum probability. $a_j$ would be predicted as the o.o.d. action

---

**Algorithm 1:** Train policy

---

**Data:** Offline Dataset $\mathcal{D}$, the action encoder $\phi$, *Any* offline RL algorithm $Y$, and number of iterations $T$.
**Result:** Policy $\pi_{i,\theta}(\hat{e}|s)$
**for** $t = 0$ *to* $T$ **do**
 Sample data $(s, a, s', r) \sim \mathcal{D}$;
 $e \leftarrow \phi(a; s)$;
 Update policy $\pi_{i,\theta}(\hat{e}|s)$ with $(s, e, s', r)$ using the given algorithm $Y$
**end**

---

**Algorithm 2:** Pseudometric-based representation learning

---

**Data:** Offline Dataset $\mathcal{D}$ and number of iterations $T$.
**Result:** Action encoder $\phi$
**for** $t = 0$ *to* $T$ **do**
 Sample data $(s, a_i, \mathcal{R}(s, a_i)) \sim \mathcal{D}$;
 Sample the other action $a_j \sim \mathcal{A}$;
 Compute representations $e_s^i = \phi(a_i; s)$ and $e_s^j = \phi(a_j; s)$;
 Estimate the transition distributions: $\hat{\mathcal{P}}(\cdot|s, e_s^i)$ and $\hat{\mathcal{P}}(\cdot|s, e_s^j)$;
 Estimate the reward of $(s, a_j)$: $\hat{\mathcal{R}}(s, e_s^j)$;
 Estimate whether $a_j$ is an o.o.d. action: $\hat{I}_\beta(a_j|s)$;
 Compute the estimated pseudometric distance: $\hat{d}(a_i, a_j|s)$ Eq.3;
 Train encoder: $J(\phi) = (\|e_s^i - e_s^j\|_1 - \hat{d}(a_i, a_j|s))^2$;
 Train transition model: $J(\hat{\mathcal{P}}, \phi) = (\hat{\mathcal{P}}(\cdot|s, e_s^i) - s')^2$;
 Train reward model: $J(\hat{\mathcal{R}}, \phi) = (\hat{\mathcal{R}}(s, e_s^i) - \mathcal{R}(s, a_i))^2$;
 Train prediction model: $J(G) = CrossEntropy(G(\cdot|s), a_i)$ ;
**end**

---

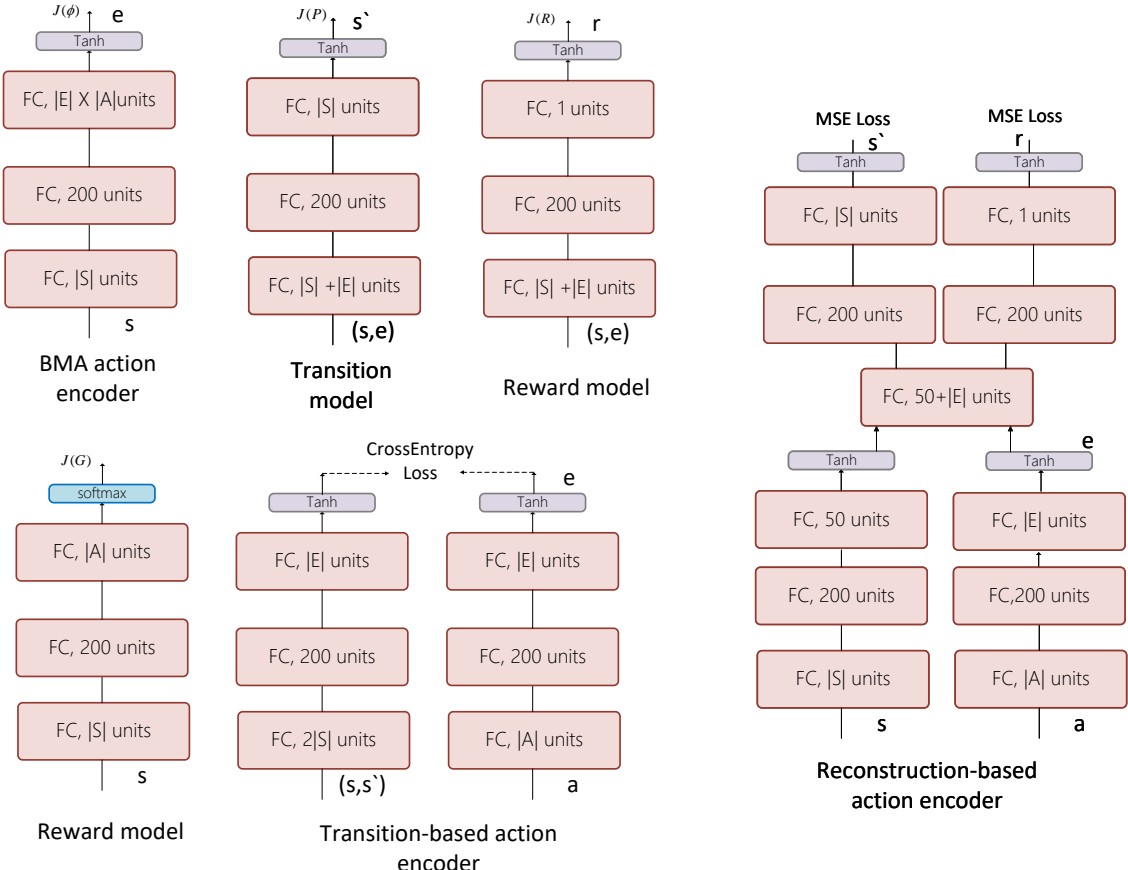

Figure 4: **Details of the network architectures used in our experiments.**

if its relative probability is below a threshold constant $\tau$. This procedure can be summarized as $\hat{I}_\beta(a|s) = \frac{G(a|s)}{\max_{\hat{a}\in\mathcal{A}} G(\hat{a}|s)} \leq \tau$ . We have searched over $\tau = \{0, 0.1, 0.3, 0.5, 0.7, 0.9, 1\}$ and found that the best setting is $\tau = 0.3$

Furthermore, the transition model $\hat{\mathcal{P}}$, the reward model $\hat{\mathcal{R}}$, and the distribution detection model $G$ are all implemented by neural networks. Their learning objective can be represented as follow:

$$
\begin{aligned}
J(\hat{\mathcal{P}}) &= (\hat{\mathcal{P}}(\cdot|s, e_s^i) - s')^2 \\
J(\hat{\mathcal{R}}) &= (\hat{\mathcal{R}}(s, e_s^i) - \mathcal{R}(s, a_i))^2 \\
J(G) &= CrossEntropy(G(\cdot|s), a_i)
\end{aligned}
\tag{24}
$$

During training, all data are randomly sampled from the offline dataset. We set the batch size as 128 and set the training gradient steps for all models as 10000. We control the scale of the learning objective function in all models by controlling the optimization procedure. It is conducted using Adam with a learning rate of $10^{-2}$, and with no momentum or weight decay. We set the dimension of the action representations $|\mathcal{E}| = 2, 2, 10, 30$ and the penalty coefficient $p = 0.01, 0.1, 0.1, 0.3$ respectively in the maze environment, the multi-step maze environment, the recommender system, and the dialogue system respectively. Details of the neural network architectures used in our experiments are provided in Fig.4 in the newest version.

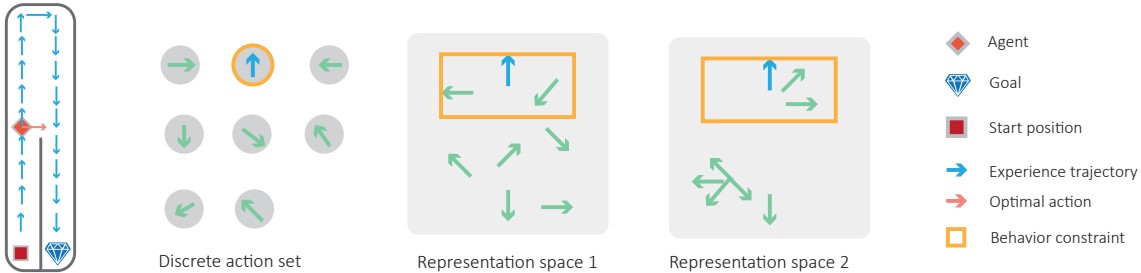

Figure 5: **An example of the influence of different action spaces on the behavioral regularizations of offline RL algorithms.**

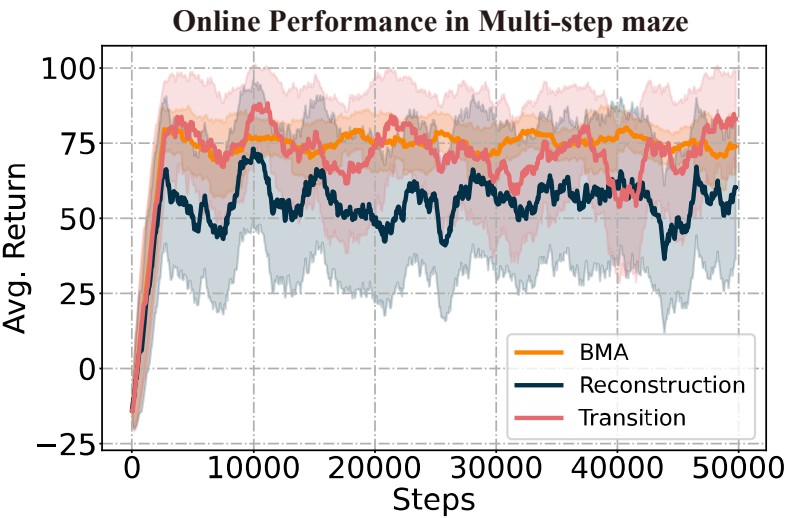

Figure 6: **Online performance of BMA, the transition-based action representations, and the reconstruction-based representations in the Multi-step maze environment.**

For the applied offline RL algorithms: BCQ (Fujimoto et al., 2019b), CQL(Kumar et al., 2020), and their discrete versions, we all adopt their open-source implementations released by the authors. For the discrete CQL, We have searched over the crucial hyperparameter $\alpha = \{0.1, 0.3, 0.5, 0.7, 0.9\}$, which determines the extent of conservative estimation of value functions. The best settings for four environments (Maze, Multi-step maze, Recommendation system, Dialogue system) are 0.5, 0.5, 0.5, 0.3, respectively. For the discrete BCQ, we have searched over $0.1, 0.3, 0.5, 0.7, 0.9$ for the hyperparameter of the threshold $\tau$, which determines the range of the candidate actions. The best settings for four environments (Maze, Multi-step maze, Recommendation system, Dialogue system) are 0.3, 0.3, 0.3, 0.3, respectively.

## A.4 ENVIRONMENTS AND DATASETS

**Maze:** We first adopt the maze environment introduced in Chandak et al. (2019). The state is the coordinates of the agent's current location, and the agent has $n$ equally spaced actuators (each actuator corresponds to a unit of movement in one direction). It can choose whether each actuator should be on or off. The effect of an action is the vectorial summation of the movements associated with the selected actuators. Thus, the size of the action space $|\mathcal{A}| = 2^n$. We set $n = 12$, so there are 4096 actions in total. There are three reward types: a reward of 10 is given if the agent visits a rewarding region, a reward of -20 is given for visiting a puddle region, and a reward

of 100 is given with the episode ending if it is on a goal region. The noise was added to the action 10% of the time, and the maximum episode length was 60 steps.

**Multi-step maze:** Selecting one decision from all possible $m$-step plans is a general problem with a large action set. If an environment has $i$ actions available at each step, the number of $m$-step plans would increase exponentially. In this setting, $|\mathcal{A}| = m^i$. We implement a version of this task on the maze environment mentioned above. The reward settings are the same in this task, but the agent only has 4 base actions: down, right, upward, and left. It needs to choose a 6-step plan per 6 time steps. Thus, the actual action size in this environment is $4^6 = 4096$.

**Recommender environment:** We consider a real-world domain with a large action set. The agent needs to recommend different videos to the simulated users, and its goal is to extend the user's viewing time. We utilize the open-source platform Recsim (Ie et al., 2019) and adopt the provided scenario called `interest_evolution` as our simulated environment. There are 1000 candidate videos in the action set. All videos are split into 20 types, and each video has its quality and video length. The sampled user has different initial interest values ranging from -1 to 1 on different video types at each episode. The user would decide whether to view the recommended video according to the interest value of the video's type. The higher the interest value, the more likely the user is to view the video. The user has a latent attribute called time budget to control whether to continue to view another video. The episode ends when the time budget is smaller than 0. The initial time budget is 100, and it would be deducted by the video length if the user has viewed a video. In addition, the viewed video's quality would also influence the user's interest value and time budget: a high-quality video brings an increase, but a low-quality video leads to a decrease. To improve the diversity of the recommended videos, we deduct the time budget by 2.5 if two videos of the same type are recommended successively. The observed state at each time step is the users' interest values and the history of its viewed videos' types. A reward equaling the video length is given once one video is viewed.

**Dialogue environment:** Building dialogue systems for providing information (Young, 2006) or improving engagement (Li et al., 2016) is a real-world challenge. One solution is to employ RL to optimize dialog strategies in multi-turn dialog models. We adopt an open-source platform called convlab (Lee et al., 2019) to simulate a dialogue environment in which each action is a dialogue. We focus on the MultiWOZ domain (Eric et al., 2019). Its main task is to help a tourist in various situations involving multiple sub-domains, such as requesting basic information about attractions and booking a hotel room. Specifically, there are 7 sub-domains - Attraction, Hospital, Police, Hotel, Restaurant, Taxi, Train. We adopt dialogues in the top 1500 using frequencies as our action set. In detail, we utilize the provided scenario called `UserPolicyAgendaMultiwoz` to simulate the tourists' policies and use `MultiWozStateEncoder` to encode the histories of dialogs into vectorized states.

**Data collection:** Since there are no open-source datasets for the offline RL tasks with large discrete action spaces, we collected logged experience trajectories generated from online RL policies. These policies were all developed by the open-source implementation of the work (Chandak et al., 2020), which can be used to address tasks with large discrete action spaces by setting the hyperparameter of the action change number as 1. We trained policies until they achieved sub-optimal performance. Then, policies of checkpoints were used to collect transition data, and the noise was added to the action 50% of the time. In this way, we collected 100000 pieces of transition data in each environment.

### A.5 EXAMPLE

To better understand the influence of action spaces to the behavioral regularization of offline algorithms, we give a toy example iluustrated in Fig.5. The task is to reach the goal from the starting position as fast as possible, and collisions with the wall induce negative reward. The agent needs to learn a policy from the logged suboptimal (blue) trajectory. Assuming that a simple behavioral regularization scenario is applied to the target policy: the agent can only select actions within a particular range near the in-distribution action. At the agent's current position, we can find that different action spaces induce different results of the behavioral regularization. The target policy is limited to selecting the sub-optimal in-distribution action (move upward) in the discrete action space since other actions are isolated from it. By contrast, the policy is permitted to choose actions near the in-distribution action in the other representation spaces where there could be some other actions in the range of behavioral regularization. This is crucial to the policy improvement since the agent has the opportunity to choose the optimal action (move right). However, actions' relative distances also largely influence the results of the behavioral regularization. In representation space 1, behaviorally different actions (the transitions and

rewards induced by them are different) are grouped together, so the agent is more likely to choose 'bad' actions. By contrast, if the relative distances between behaviorally similar actions are close (like in representation space 2), the agent tends to select the optimal or other 'safe' actions.

## A.6 ONLINE PERFORMANCE OF BMA

Actually, BMA action representations can be directly applied on the online setting. Utilizing the framework mentioned in Chandak et al. (2019) which can incorporating action representaions and online RL algorithms. We get the online performance of BMA, the transion-based representations, and the reconstruction-based representations in the Multi-step maze environment. The results are illustrated in Fig.6. We can find that, in the online setting, three action representations can all achieve fast convergence, while only BMA can get good performance in the offline setting (Fig.2 (b)). This indicates that the BMA is more effective in offline RL tasks with large discrete action spaces.

