# OpenReview forum: "Learning Pseudometric-based Action Representations for Offline Reinforcement Learning"
_ICLR.cc/2022/Conference — ICLR 2022 Submitted_

### Official Review · Reviewer_pJJJ · 2021-11-01

**Correctness:** 3
**Technical Novelty And Significance:** 1
**Empirical Novelty And Significance:** 2
**Recommendation:** 5
**Confidence:** 4

**Main Review:**

Strength:
  - This paper is well-written to follow. It clearly explains the problem, the motivation, and the proposed method.
  - It provides theorems on the continuity and the generalization of Q-function.
  - The proposed algorithm on environments with large discrete action spaces can improve other offline algorithms' performance, and the algorithm using the learned representation space outperforms other representations.

Weakness:
  - The proposed BMA is a simple extension of the bisimulation metric [1, 2] for offline RL. The pseudo-metric is a bisimulation metric for actions plus an indicator function that outputs one when the two actions are drawn from the identical distributions and 0 otherwise.
  The minor thing is that the explanation of Theorem 2 is too similar to that of Theorem 2 of [2]. I think that the explanation should be rewritten.

Typo:
  - In eq (8), the reward difference term
  - In the 5.1 Experimental results section on page 8, "annotated by BMA-CQL and BMA-CQL, respectively."

[1] N. Ferns, P. Panangaden, and D. Precup. Metrics for finite Markov decision processes. In UAI, 2004.

[2] A. Zhang, R. McAllister, R. Calandra, Y. Gal, and S. Levine. Learning invariant representations for reinforcement learning without reconstruction.


**Summary Of The Paper:**

This paper proposed a pseudo-metric between actions, called Behavioral Metric of Actions (BMA), for offline RL in environments with large discrete action spaces. This metric considers both the data-distributional relation and the behavior relation. The distance between two actions drawn from similar distributions with similar rewards and transition probabilities is small. This paper also provides theorems about the continuity and generalization of Q-function using the proposed pseudo-metric. The experimental results show that training a policy in the proposed action representation space has significantly improved performance for offline RL algorithms in environments with large discrete action spaces.

**Summary Of The Review:**

This paper has interesting experimental results, but the proposed method and the theory are not novel enough to accept.

---

> ### Author Response · Authors · 2021-11-22
> **Response to Reviewer pJJJ**
>
> We thank the reviewer for your helpful feedback. In the following response we will explain why a direct extension of bisimulation metric does not work in offline RL. In addition, we derive and explain a new theorem to demonstrate the theoretical policy improvement under BMA (Please refer to the rewritten Theorem 2 in the new version). We also highlighted the novelties of our work in the general response. We would greatly appreciate if the reviewer could spend some time on reading the new version of our paper and reevaluate our contributions. Thank you!
>
> **Q1. The proposed BMA is a simple extension of the bisimulation metric for offline RL.**
>
> A1. Although the bisimulation metric has been studied in learning state representations. However, we argue that it is not straightforward to extend it to learn action representations, since “state” and “action” are intrinsically different in the context of MDP. Specifically, we made following three key contributions to make BMA work in offline RL.
>
> **Regarding the formulation, BMA is a conditional metric**. Specifically, we refer to the bisimulation metric of state as Eq.(1) [2]:
> $$
> d(s_i,s_j) = \max_{a \in \mathcal{A}}|\mathcal{R}(s_i,a)-\mathcal{R}(s_j,a)| + \gamma \cdot W_2(\mathcal{P}(s'|s_i,a),\mathcal{P}(s'|s_j,a))  \quad (1)
> $$
>
> If we directly convert Eq.(1) to the metric on the action space, we get Eq.(2) as follow:
> $$
> d(a_i,a_j) = \max_{s \in \mathcal{S}}|\mathcal{R}(s,a_i)-\mathcal{R}(s,a_j)| + \gamma \cdot W_2(\mathcal{P}(s'|s,a_i),\mathcal{P}(s'|s,a_j))  \quad (2)
> $$
> This simple extension needs to search over the entire state space to find the max value. Therefore, it is not tractable when the state space is large or even continuous, which is quite common in practice.
>
> To address the issues mentioned above, we set BMA as a state-conditioned metric of actions as follows:
> $$
> d(a_i,a_j|s) = |\mathcal{R}(s,a_i)-\mathcal{R}(s,a_j)| + \gamma \cdot W_2(\mathcal{P}(s'|s,a_i),\mathcal{P}(s'|s,a_j))  \quad (3)
> $$
>
> This improvement brings three key advantages, which make BMA a suitable metric for offline RL. (1) It does not require to search over the entire state space. (2) It brings tighter bounds in Theorem 1 and Theorem 2. (3) The metric becomes policy-agnostic, which means that BMA is stable even when the data is collected from different behavior policies.
>
> **Regarding the learning method, we propose a novel architecture to overcome the data-poor problem.** For each state, the dataset only records the action that actually taken at that state. Therefore, we do not know the effect of the majority of the actions. To overcome this data-poor problem, we propose to use a relation network architecture (Figure 1b) that balance between in-distribution and out-of-distribution actions. Note that this problem does not exist in [1].
>
> **Regarding the theoretical contributions.** We have followed the reviewer’s suggestions to rewrite Theorem 2. Furthermore, we derived a bound of the difference between value functions in the original MDP and the constructed MDP (Please refer to Theorem 2 in the new version. The original Theorem 2 is moved to Appendix due to space limit).
>
> We proved that with a high probability, learning an offline policy in the constructed pseudometric-based MDP would have a better performance compared with learning policy in the original MDP. This policy improvement results from two factors: the reduction of the dimension of action spaces and the learning error of the action encoder. The first factor can be easily controlled, since we can set $|\mathcal{E}| \ll |\mathcal{A}|$, where $|\mathcal{E}|$ is a hyperparameter (the dimension of the action representation space). The second factor is the learning error induced by a self-supervised loss function. Compared with errors induced by offline RL algorithms, this term is much easier to be controlled. In conclusion, BMA provides a theoretically guarantee of policy improvement for offline RL tasks with large discrete action spaces.
>
> [1] A. Zhang, R. McAllister, R. Calandra, Y. Gal, and S. Levine. Learning invariant representations for reinforcement learning without reconstruction. In ICLR, 2021.

---

> > ### Comment · Reviewer_pJJJ · 2021-11-29
> > **Post Rebuttal**
> >
> > Thanks for the response. I've read other reviewers' comments and the authors' responses carefully. I agree with the authors that the paper is the first approach to solve offline RL with large discrete action space using the state-conditioned bisimulation metric for actions. However, I still think that the paper's novelty of the main idea is not significant.
> >
> > 1. The most interesting part of the response is that the proposed method outperforms the previous method using a bisimulation metric of state-action pairs [1]. However, I also think that the explanation of this result is a conjecture, as mentioned by the reviewer hprr.
> >
> > 2. There is a critical error in the statement of Theorem 2 in the new version of this paper. First of all, the authors CANNOT replace $|\mathcal{A}|$ with $|\mathcal{E}|$. The theorem 3.6 of [2], which is the original version of (22) in page 13, holds only when the action space is finite. In the derivation of theorem 3.6 in [2], it used Cauchy-Schwarz inequality ($D_{CQL}(s) = \sum_a \frac{\pi(a|s)^2}{\hat{\pi}_\beta(a|s)} - 1 \Rightarrow D_{CQL}(s) + 1 \leq \left( \sum_a \frac{\pi(a|s)}{\sqrt{\hat{\pi}_\beta(a|s)}} \right)^2 \leq |\mathcal{A}| (D_{CQL}(s) + 1) $),
> >
> > and this means that the inequality holds when the action space is finite. However, the authors used the dimension of a continuous representation space $|\mathcal{E}|$ instead of the number of actions $|\mathcal{A}|$, and this is wrong. Therefore, the inequality does not depend on $|\mathcal{E}|$ and that means that the theorem does not provide the effectiveness of learning on a action representation space.
> >
> > Although I raised my score because of some experimental results, concerns about the novelty of the main idea remain. In addition, the authors should consider the mentioned errors in the theoretical statements.
> >
> > [1] R. Dadashi, S. Rezaeifar, N. Vieillard, L. Hussenot, O. Pietquin, and M. Geist. Offline reinforcement learning with pseudometric learning, ICML 2021.
> >
> > [2] A. Kumar, A. Zhou, G. Tucker, S. Levine. Conservative Q-learning for offline reinforcement learning, NeurIPS 2020.

---

> > > ### Author Response · Authors · 2021-11-30
> > > **Response to Reviewer pJJJ: Baseline & Theorem 2**
> > >
> > > Thank you for reading our new version and reevaluating our work. We are very grateful.
> > >
> > > **Q1. Experimental results compared with [1].**
> > >
> > > A1. As is raised by Reviewer hprr, it is interesting to compare BMA with other bisimulation based metrics. However, the metric in [1] is used to contrain the policy, while our metric is used to learn action representations. In other words, the motivation of the two metrics are different.  Therefore, we believe that **Transition**, **Reconstruction**, **External** and **PLAS** are more proper baselines to evaluate BMA, as reported in the paper.
> > >
> > > **Q2. About Theorem 2.**
> > >
> > > A2. In fact, **the number of latent actions is still finite because the original action space is finite**. Therefore, Theorem 2 still holds. We relax the latent action space to be continuous for the ease of optimization using gradient descent. By this way, $|\mathcal{E}|$ is the number of projected latent actions instead of the dimension of the latent action space. In general, we have $|\mathcal{E}| \le |\mathcal{A}|$. However, in special cases where actions naturally group together (perhaps under some regularizations), we will have $|\mathcal{E}| \ll |\mathcal{A}|$.
> > > For example, in the Maze environment, the original actions correspond to the combined state of 8 atuators, and $|\mathcal{A}| = 2^8 =4096$. However, since the actual effect of an action is the vectorial summation of the displacements associated with the selected actuators, there are only 722 unique efects of actions ($|\mathcal{E}| = 722$). This also gives us an intuition: **BMA provides a tighter bound for policy improvement when the actions group together in the latent spaces.**
> > >
> > > Thanks again for reevaluating our work. Please let us know if the above explanations do not address your concern or you have further questions. Thank you!
> > >
> > > [1] R. Dadashi, S. Rezaeifar, N. Vieillard, L. Hussenot, O. Pietquin, and M. Geist. Offline reinforcement learning with pseudometric learning, ICML 2021.

---

### Official Review · Reviewer_hprr · 2021-11-02

**Correctness:** 4
**Technical Novelty And Significance:** 3
**Empirical Novelty And Significance:** 2
**Recommendation:** 6
**Confidence:** 2

**Main Review:**

Strength:

I think the paper is well-written and easy to follow, the motivation of learning a representation specifically for offline RL learning is very clear. The empirical evaluations show good improvement of the proposed method than other existing ones.

Weakness:

I have the following concerns/questions
1. Since [1] is also working on using pseudometric for representation as mentioned in Section 2, it would be a more proper baseline to compare with. Besides, I'm particularly interested in this comparison because I think one of the difference between [1] and BMA is that the distance defined in BMA is conditional on the states, while the distance in [1] is defined jointly by states and actions. There has been discussions about the conditional distribution and joint distribution in other areas such as imitation learning [2], so I'm interested to see the comparisons.
2. The hyparameters of the alternatives in Figure 5 are all default parameters from their prior settings, which make the comparisons less convincing since the tasks are very different. It would be more convincing if a grid search over the hyperparameters as indicated in their original papers can be applied.
3. It would be also interesting to see how the performances will be changed with respect to the value of gamma since the transition distributions can be challenging to estimate when the action dimension is large.

Minor point:

1. Eq(6) looks confusing because $||e_s^i - e_s^j||$ and $d(a_i,a_j|s)$ are the same by definition.
2. Will the complexity of eq(1) be a problem when the action space is very large? (e.g., on recommendation system situation, the action space can be million scale)

[1]: R. Dadashi, S. Rezaeifar, N. Vieillard, L. Hussenot, O. Pietquin, and M. Geist. Offline reinforcement learning
with pseudometric learning, 2021.
[2] Ghasemipour, Seyed Kamyar Seyed, Richard Zemel, and Shixiang Gu. "A divergence minimization perspective on imitation learning methods." Conference on Robot Learning. PMLR, 2020.

**Summary Of The Paper:**

This paper proposes a framework for learning Behavioral Metrics of Actions that combines both behavioral relation and data-distribution relation between actions to help the learning of offline RL algorithms on large discrete-action tasks.


**Summary Of The Review:**

In summary, I think the proposed method BMA is interesting and the evaluation is promising. It would be great to add a few experiments as mentioned above to make more fair comparisons and obtain a more convincing conclusion.

---

> ### Author Response · Authors · 2021-11-22
> **Response to Reviewer hprr**
>
> Thanks for your helpful feedback and kind suggestions! Please see our responses bellow.
>
> **Q1. Since [1] is also working on using pseudometric for representation as mentioned in Section 2, it would be a more proper baseline to compare with.**
>
> A1. Although both [1] and BMA use pseudometric for representation learning, the defined pseudometrics are quite different. The metric proposed by [1] is to measure the behavioral relation between **joint** state-action pairs, while BMA is a metric to measure the relations between actions **conditioned** on the given state. The drawback of defining metric on joint state-action pairs is that the state information may dominate the action information, especially in the setting of large action space where the influence of each individual action is weak. To compare [1] with BMA, we adopt the pseudometric proposed in [1] and carried experiments in four environments (refer to Section 5 for detailed descriptions). To be fair, we trained [1] with 50000 gradient steps and carried 4 independent run of different seeds. We reported the maximum results of [1] and BMA as follows:
>
> |   |  Maze | Multi-step maze  | Recommendation system  | Dialogue system   |
> | ------------ | ------------ | ------------ | ------------ | ------------ |
> |  [1] |  $0.0 \pm 0.0$ | $32.3 \pm 15.1$  | $51.4 \pm 1.2$  | $0.2 \pm 0.1$  |
> | BMA  |   $96.0 \pm 2.1$ |  $92.3 \pm 2.0$  |  $98.1 \pm 5.2$  |  $0.6 \pm 0.1$  |
>
> We can find that [1] cannot show effective performance in tasks with large action spaces, which indicates that the metric proposed in BMA is more suitable for tasks with large action spaces.
>
> **Q2. The hyparameters of the alternatives in Figure 5 are all default parameters from their prior setting**
>
> A2. We have conducted additional experiments to search the best values for crucial parameters of baselines in Figure 5. For the discrete CQL, we have searched over ${0.1, 0.3, 0.5, 0.7, 0.9}$ for the parameter of $\alpha$, which determines the extent of conservative estimation of value functions (The default $\alpha$ is ). The best settings for four environments (Maze, Multi-step maze, Recommendation system, Dialogue system) are 0.5, 0.5, 0.5, 0.3. We have updated Fig.1 in the newest version.
>
> For the discrete BCQ, we have searched over ${0.1, 0.3, 0.5, 0.7, 0.9}$ for the parameter of the threshold, which determines the range of the regularized actions (The default threshold is 0.3). The best settings for four environments (Maze, Multi-step maze, Recommendation system, Dialogue system) are 0.3, 0.3, 0.3, 0.3. The default parameter leads to the best results. So, we didn't update relevant results of the discrete BCQ.
>
> **Q3. It would be also interesting to see how the performances will be changed with respect to the value of gamma**
>
> A3. We report the results of different gammas as follows, and will also add these results in Appendix.
>
> |   |  $\gamma = 0$ | $\gamma=0.3$  | $\gamma=0.6$   | $\gamma=0.9$   |
> | ------------ | ------------ | ------------ | ------------ | ------------ |
> | Maze  | 0.0 | 32.3  | 90.0  | 96.0  |
> | Multi-step maze  | 6.2  |  90.4 | 77.1  | 92.3  |
> | Recommendation system  | 42.1  | 75.9  | 98.5  |  98.1 |
> | Dialogue system  |  0.1 | 0.4  | 0.6  | 0.6  |
>
> We can find that when $\gamma=0$, BMA shows poor performance. This is reasonable, since in this setting, BMA cannot capture any information about transitions. When $\gamma \ge 0$, the performance of different $\gamma$s is varing in a reasonable range. This motivates us to search for the best $\gamma$ before deployment.
>
> **Q4. Eq(6) looks confusing.**
>
> A4. $\|e_s^i-e_s^j\|_1$ indicates the $\ell_1$ distance between two action representations, while $d(a_i,a_j|s)$ indicates the distance computed by our metric function.
>
> **Q5. Time complexity of Eq.1.**
> A5. Actually，Eq.1 equals to finding the minimum value among $N$ actions, so its time complexity is $O(N)$, which means that it can be computed efficiently.
>
>
> [1] R. Dadashi, S. Rezaeifar, N. Vieillard, L. Hussenot, O. Pietquin, and M. Geist. Offline reinforcement learning with pseudometric learning, 2021.

---

> > ### Comment · Reviewer_hprr · 2021-11-22
> > **Update**
> >
> > Thank you for the response. The additional experiments in general make me convincing that the proposed method BMA does make improvement compared with fair baselines. Though I'm still skeptical about the *drawback* of using distribution of joint state-action pair, the experiments demonstrate BMA outperforms [1] with a clear margin. I thereby incline to raise my score.

---

> > > ### Author Response · Authors · 2021-11-30
> > > **Response to Reviewer hprr**
> > >
> > > Thanks for the update.
> > >
> > > To make it more clear, we think that the metric proposed in [1] cares the distance between state-action pairs, and thus suitable for constraining the policy to stay clase to dataset. By contrast, BMA cares the distance between pure actions, and thus more suitable for learning action representations. Hope it helps. Thank you!
> > >
> > > [1] R. Dadashi, S. Rezaeifar, N. Vieillard, L. Hussenot, O. Pietquin, and M. Geist. Offline reinforcement learning with pseudometric learning, 2021.

---

### Official Review · Reviewer_bL2q · 2021-11-02

**Correctness:** 3
**Technical Novelty And Significance:** 2
**Empirical Novelty And Significance:** 2
**Recommendation:** 5
**Confidence:** 3

**Main Review:**

Strength:
- I like the setting this paper studies, how to learn efficient action representation, which seems a necessary task when we face large discrete action spaces, which is also the case for some real-world applications.
- The paper is well-written and it is easy to follow.
- The experiments are well done. I do like the ablation part, which demonstrates the effectiveness and necessarity of different components.

Weakness:
- The novelty of this paper seems very limited. For the behavior metric, it just seems like doing some bisimulation metric under the action space, while there are already tons of literature studies on this over the state space. This is a simple and trivial extension to the action space.
- Regarding the behavior metric, it would be great if you can also the effectiveness in the online setting since this is not a component specific to offline RL.
- Specific to offline RL, it adds a distributional penalty. Though some theoretical results are given, I do not gain much insight on it regarding how this distributional penalty helps there. Theorem 1 and 2 seem also to hold without this distributional term in the metric. How this helps theoretically, compared with not adding it.
- Also for Figure 7, is there any explanation that the metric will give more compact clustering than the reconstruction-based method, and why is this the case?



**Summary Of The Paper:**

This paper studies the action representation learning for offline RL with large discrete action spaces. It proposes a new metric to capture the similarity between actions, called Behavioral Metrics of Actions. The new metric has two components, one is a behavioral metric that groups actions that share the similar MDP property (i.e., reward and transition model), and the other offline specific one is distributional metric, which uses a binary variable to capture whether the two actions coming from the same data distribution. Empirical studies show superior performance over the baselines, and the ablation study demonstrates the necessity of all of the components in the metric.

RL with large discrete action spaces is of interest in some real-world applications, such as recommender systems and dialogue systems, this paper gives a way to do the action abstraction to further reduce the action space and hence increase the effective sample size in any downstream learning task.

**Summary Of The Review:**

This paper is well-written and the experiments are well-done to illustrate the effectiveness of the proposed metric. However, the novelty of this paper is pretty low, as a bisimulation type metric is well known. Though adding a new distributional component in the offline setting helps, it does not well-justified at all. Given this, I recommend a weak reject for the paper.

---

> ### Author Response · Authors · 2021-11-22
> **Response to Reviewer bL2q**
>
> We thank the reviewer for your useful feedback. Hope that our responses help to clear your concerns.
>
> **Q1. The novelty of this paper seems very limited.**
>
> A1. We have highlighted the novelty of our work from different perspectives. Please refer to the shared response for details. We greatly appreciate that you could reevaluate the novelty of our work depending on our response.
>
> **Q2. It would be great if you can also examine the effectiveness in the online setting.**
>
> A2. Although BMA is designed for the offline setting, it can also be implemented in an off-policy manner in the online setting. We have carried additional experiments to compare BMA with two baselines (**transition** and **reconstruction**) in the maze environment. The experimental results show that BMA performs similar to “transition” and better than “reconstruction”. This indicates that the metric defined in BMA also helps the generalization between actions in online setting. However, in the offline setting, BMA shows significant advantage over the two baselines. This supports the core hypothesis of our work: **A reasonable and explicitly defined metric between actions is key to escalate offline RL with large action spaces**. The intuition is that offline RL is sensitive to o.o.d. actions, and a stable metric helps to regularize the learning from such actions.
>
> **Q3. Though some theoretical results are given, I do not gain much insight on it regarding how this distributional penalty helps there.**
>
> A3. We have added some explanations on the influence of the distributional penalty. In one word, it influences the upper bound on the differences of two action values when they are from different distributions. Specifically, recall that we have $|Q^{\pi}(s,a_i) - Q^{\pi}(s,a_j)| \le \frac{1}{1-\gamma} \cdot d(a_i,a_j|s)$ by theorem 1. This conclusion can be further explained as: If one of $a_i$ and $a_j$ is the o.o.d. action but the other is not, we have $|Q^{\pi}(s,a_i) - Q^{\pi}(s,a_j)| \le \frac{1}{1-\gamma} \cdot(|\mathcal{R}(s,a_i)-\mathcal{R}(s,a_j)| + \gamma \cdot W_2(\mathcal{P}(s’|s,a_i),\mathcal{P}(s’|s,a_j) + p)$ Otherwise, we have $|Q^{\pi}(s,a_i) - Q^{\pi}(s,a_j)| \le \frac{1}{1-\gamma} \cdot(|\mathcal{R}(s,a_i)-\mathcal{R}(s,a_j)| + \gamma \cdot W_2(\mathcal{P}(s’|s,a_i),\mathcal{P}(s’|s,a_j))$. This indicates that if two actions are from different distributions, the difference between their values tend to have a looser upper bound depending on $p$, and $p$ plays an important role in balancing between penalizing and exploring the o.o.d. actions. If $p$ is too large, the difference between two values tends to be large, and the offline RL algorithms would ignore most o.o.d. actions. If $p$ is too small, the algorithms would explore too many o.o.d. actions and incur large extrapolation error.
>
>
> **Q4. Also for Figure 7, is there any explanation that the metric will give more compact clustering than the reconstruction-based method, and why is this the case?**
>
> A4. The loss function ($J(\phi) = \left( \|e_s^{i} -e_s^{j}\|_1 - d(a_i,a_j|s)\right)^2$) in our learning framework introduces an explicit mechanism to aggregate the action representations with similar behaviroal effects. As a result, if two actions have similar effect, they would be forced to stay close in the representation space. This regularization leads to a more compact representation structure. By contrast, the reconstruction-based baseline has no mechanism to regulate the distances between actions, so that they could diverge in the representation space, possibly due to some noise in data.

---

### Official Review · Reviewer_gR99 · 2021-11-03

**Correctness:** 4
**Technical Novelty And Significance:** 3
**Empirical Novelty And Significance:** 3
**Recommendation:** 6
**Confidence:** 3

**Main Review:**

## Goals
This paper sets out to define a new action metric for large action spaces which aids in learning on both on-line and off-line scenarios.  They provide a reasonable approach that is demonstrated to perform empirically.

## Strengths
* Generalization over states has been one of the main powerful aspects of function-approximated RL, however as actions are often without any direct feature representation they have often been treated as a an arbitrary discrete set.  Providing representations over actions is key to scaling RL up to large systems, and is therefore an important problem to tackle.

* The proposed approach makes good intuitive sense, and the conclusion that the embedding line up to correlate with the value function is an important one, as this has been my personal intuition for a while: as far as the policy is concerned with optimizing the return, two actions are similar iff they have the same expected value.

* The ablations are clear, especially around the value of p which is nice to see.

* The embedding visualizations are nice on the maze, however it's a bit surprising that the embedding from Chanak don't look better as they have figures in their paper which seem more well-distributed for a maze environment.

* Overall the bounds make sense and are intuitive, and looking over the proof the general idea seems good, I am however not the right person to give them a true seal of approval.

## Weaknesses
The paper has a couple important issues, more in the form than the function:
* Many figures are small, with very small text for the legends, pleas try and make the text bigger and consume more surrounding whitespace with your figures as much as possible.

* There is a lot of space spent on presenting the problem, not leaving enough for clear experimental results and discussion.  Perhaps it's because I'm already convinced of the importance of this problem, but I think for example that Figure 2. doesn't bring much to the table and actually confused me more than anything at first.

* Implementation details are sorely lacking: what is your nearest-neighbour lookup? How do do you actually set up your embedding network? I'm not entirely sure I'd be able to replicate this, please provide more concrete information on the actual configuration of the network and any tricks to get the loss to actually converge.  For example in Algorithm 2 in the appendix you mention estimating the transition distributions, but then you train them afterwards, I'm sure this makes sense somehow but it's not quite clear to me and I would appreciate it being clearer :)


## Paper Readability
Overall the approach is legible but there are some concerns that I would like addressed:
* There are no descriptions for Figure 5 and Figure 6, please provide a descriptive legend.
* All figures have very small text and have a lot of surrounding whitespace, they are overall hard to read and I had to zoom in a lot to try and understand them.
* Overall I would suggest perhaps restructuring Sec. 4.2 some to be easier to follow.
* One small NIT is that you talk about data-distribution separability, but there is really only one distribution (the expert's), and then you talk about OOD detection.  I believe what you're trying to do is OOD detection through some mechanism, whereas distrubition separability seems like there is perhaps two sources of training data.  Is there a reason not to just call this OOD detection throughout? This would reduce some confusion I had.
* For the OOD detection can you elaborate in just one sentence what Fujimoto et al.'s approcah is for I_B?  I didn't have time to go through that paper to see what they did and a one-liner would have made things clearer.

Thanks again for this work and hopefully we can clear up some of the form-related nits!


**Summary Of The Paper:**

This paper presents a novel framework for learning action metrics from offline data called Behavioral Metric of Actions (BMA).  The goal of this metric is to position the actions in a latent space that is pertinent relative to the task being learned.  This is particularly useful in environments with a large number of actions where exhaustive exploration is unfeasible, and therefore generalization over actions is necessary.  BMA has two design principles: behavioral similarity between actions, and data-distributional.  Behavioral reflects the fact that two actions that have a similar induced transition with a similar reward should be considered relatively equivalent.  The data-distributional relationship separates actions that appear within the offline dataset from ones that are absent (or at least absent for a particular state).  The authors show that by learning an embedding with these principles they are able to train off-the-shelf algorithms on offline RL tasks in cases where learning would be impossible without an action representation.  They also show that compared to other action representations their approach performs better.

**Summary Of The Review:**

I believe this paper should be accepted on the merit of having:
* A pertinent problem: action embeddings for large action spaces.
* A well thought-out approach to tackle this problem: Using a set of reasonable priors on action similarity to provide a general embedding for actions given an existing dataset of behavior on the environment.
* Convincing experimental results: Both in the on-line and off-line case the approach works well, and is shown to work better than other action embedding schemes.

However it has issues in its form:
* Clearly missing description on Figures 5 and 6.
* Hard-to-read figures for results.
* Not always easy to follow explanation of the method.

If the authors can manage to clear up at least most of the form issues I will support strong acceptance.

---

> ### Author Response · Authors · 2021-11-22
> **Response to Reviewer gR99**
>
> We thank the reviewer for your constructive feedbacks! We hope that the bellow reponses help to clear your concerns.
>
> **Q1. It's a bit surprising that the embedding from Chanak don't look better as they have figures in their paper which seem more well-distributed for a maze environment.**
>
> A1. The reason is that the illustration of embeddings in our paper is different from that in Chanak's. In our paper, the color of each embedding corresponds to its value under the given state, while the embedding color in Chanak's correponds to the displacement in the Cartesian co-ordinates caused by each action, e.g. if an action caused a displacement $(x,y)$, its corresponding color is $[R = x, G = y, B=0.5]$.
>
> **Q2. Implementation details.**
>
> A2. We will explain some of the details as follows and update in the paper accordingly. We also plan to publish our code in the near future.
>
> For the nearest neighbor lookup function $g$ can be regarded as finding the minimum distance between $\hat{e}$ and all actions' embeddings. So, we directly adopted the torch.min() function in the PyTorch python package. Its time complexity is $O(N)$, so there would not be a serious scale problem when the action set is enormous.
>
> In our implementations, the transition model $\mathcal{\hat P}$, the reward model $\mathcal{\hat R}$, and the distribution detection model $G$ are all implemented by neural networks. Their learning objective can be represented as follows:
>
> $$J(\mathcal{\hat P},\phi) = (\mathcal{\hat P}(\cdot|s,e_s^i) - s')^2$$
>
> $$J(\mathcal{\hat R},\phi) = (\mathcal{\hat R}(s,e_s^i) - \mathcal{R}(s,a_i))^2$$
>
> $$J(G) = CrossEntropy(G(\cdot|s),a_i)$$
>
> During training, all data are randomly sampled from the offline dataset, and we set the batch size as 128 and set the training gradient steps for all models as 10000. We control the scale of the learning objective function in all models by controlling the optimization procedure. It is conducted using Adam with a learning rate of $10^{-2}$, and with no momentum orweight decay. We set the dimension of the action representations $|\mathcal{E}| = 2, 2, 10, 30$ and the penalty coefficient $p = 0.01, 0.1, 0.1, 0.3$ respectively in the maze environment, the multi-step maze environment, the recommender system, and the dialogue system respectively.
> **Details of the neural network architectures used in our experiments are provided in Fig.4 in the new version.** And we have added these implementation details into the new version.
>
> **Q3. For the OOD detection can you elaborate in just one sentence what Fujimoto et al.'s approach is for I_B?**
>
> The approach first trains a model to predict $\pi_{\beta}(a|s)$. Given a state action pair $(s,a)$ and a hyperparameter $\tau$, if $\frac{\pi_{\beta}(s,a)}{\max_{a \in \mathcal{A}}\pi_{\beta}(s,a)} \le \tau$, $a$ would be regarded as an o.o.d. action. We will explain more clear in the paper.
>
> **Q4. Other issues.**
>
> 1. **Many figures are small.**
>
>     For all figures, we have made the text bigger and consumed more whitespace.
>
> 2. **There is a lot of space spent on presenting the problem, not leaving enough for clear experimental results and discussion.**
>
>     We have deleted some redundant description in Section 3 and moved Fig.2 to the appendix.
>
> 3. **There are no descriptions for Figure 5 and Figure 6, please provide a descriptive legend.**
>
>     We have added a descriptive legend for Fig.5 and Fig.6.
>
> 4. **Overall I would suggest perhaps restructuring Sec. 4.2 some to be easier to follow.**
>
>     We have restructured Sec. 4.2 to make it easier to follow.
>
> 5. **One small NIT is that you talk about data-distribution separability.**
>
>     We have changed data-distribution separability to data-distribution detection. Thanks for the suggestion.

---

### Author Response · Authors · 2021-11-22
**General Response: Novelties and Updates**

We thank all the reviewers for their time spending on our paper and insightful feedbacks. We would like to highlight the contributions and novelty of our work as follows. We sincerely hope that the reviewers could spend some time reading the new version of our paper and reevaluate our contributions.

1. **Novelty of problem setting.** Existing offline RL works focus on continuous control tasks and computer games with relatively small action space. However, industrial applications (e.g., recommender systems and dialogue systems) usually come with very large action space. **To our knowledge, we are the first to study offline RL with large and discrete action spaces.**

2. **Novelty of the BMA metric.** Bisimulation metric has been demonstrated effective in learning state representations. But whether it is suitable for learning action representations remains unclear, because state and action are intrinsically different in the context of MDP. In this work, we extend the bisimulation metric to be a conditional metric (Please refer to our response to Reviewer pJJJ for more details) and propose to use a distributional penalty to reduce the influence of o.o.d actions. We did extensive ablation experiments to demonstrate the effectiveness of BMA. Furthermore, we provide theoretically guarantees of policy improvement for offline RL tasks with large action spaces (Please refer to Theorem 2 in the new version). **To our knowledge, we are the first to demonstrate the effectiveness of the bisimulation metric in action space, both empirically and theoretically.**

3. **Novelty of learning framework.** For each state, the dataset only records the action that actually taken at that state. Therefore, we do not know the effect of the majority of the actions. To overcome this data-poor problem, we propose to use a relation network architecture (Figure 1b) that balance between in-distribution and out-of-distribution actions. Note that this problem does not exist in [1].

The updates of the new version are listed as follows. Hope that it helps the reviewers with reading the new version.

1. We have improved all figures and corresponding legends for better reading experience. In detail. We have merged Fig.1 and Fig.3 in the previous version into the new Fig.1 in the newest version, and merged Fig.4, Fig.5, and Fig.6 into the new Fig.3. To prevent confusion, we have moved the old Fig.2 to the appendix.

2. To make the paper easier to follow, we have deleted some redundant description of presenting the problem and restructured the flow in Section 4.

3. We have provided more implementation details about how to train models used in our experiments, and added a new figure (Fig.4) to desrcibe the details of network architectures.

4. We have rewritten theorem 2 in the previous version to provide a bound about the divergence beween the value function in the original MDP and that in the constructed MDP. This divergence depends on the learning erorr of BMA action encoder. For simplicity, we have moved this theorem to the appendix (annotated as lemma 1).

5. We have provided a new theorem (annotated as theorem 2 in the newest version) to prove that learning an offline policy in the BMA action representation space are easier to achieve better performance than directly learning an offline policy in the original action space.

6. We have reported the online performance of different action representations (BMA, transition-based, reconstruction-based) in Fig.6.

7. We have modified some typos.


[1] A. Zhang, R. McAllister, R. Calandra, Y. Gal, and S. Levine. Learning invariant representations for reinforcement learning without reconstruction. In ICLR, 2021.

---

### Decision · Program_Chairs · 2022-01-20

**Decision:**

Reject

**Comment:**

The paper proposes a new pseudometric for action representations.  The reviewers generally liked the work and the rebuttal helped to clarity may concerns.  However, the degree of novelty of the approach remains a concern.  In addition, a technical error was discovered by a reviewer in the revised paper.  Hence the paper is not ready for publication.